# Structural basis of human NOX5 activation

Chenxi Cui [1,4], Meiqin Jiang[1,4], Nikhil Jain[1,4], Sourav Das [2], Yu-Hua Lo[1], Ali A. Kermani[1], Tanadet Pipatpolkai [3]✉ & Ji Sun [1]✉

NADPH oxidase 5 (NOX5) catalyzes the production of superoxide free radicals and regulates physiological processes from sperm motility to cardiac rhythm. Overexpression of NOX5 leads to cancers, diabetes, and cardiovascular diseases. NOX5 is activated by intracellular calcium signaling, but the underlying molecular mechanism of which − in particular, how calcium triggers electron transfer from NADPH to FAD − is still unclear. Here we capture motions of full-length human NOX5 upon calcium binding using single-particle cryogenic electron microscopy (cryo-EM). By combining biochemistry, mutagenesis analyses, and molecular dynamics (MD) simulations, we decode the molecular basis of NOX5 activation and electron transfer. We find that calcium binding to the EF-hand domain increases NADPH dynamics, permitting electron transfer between NADPH and FAD and superoxide production. Our structural findings also uncover a zinc-binding motif that is important for NOX5 stability and enzymatic activity, revealing modulation mechanisms of reactive oxygen species (ROS) production.

Reactive oxygen species (ROS) are involved in various critical biological processes, and maintaining ROS homeostasis is essential for human health[1]. NADPH oxidase 5 (NOX5) mediates ROS production and contributes to normal physiology and disease pathogenesis[2]. NOX5 is widely expressed in different tissues and cell types, including spermatocytes, lymph nodes, spleen, kidney, oligodendrocytes, and cardiomyocytes, and regulates essential physiological processes like sperm motility, cell proliferation, and host defense[3–6]. Overexpression of NOX5 is associated with numerous human diseases, including cardiovascular diseases, nicotine-related atherosclerosis, renal injury, diabetic nephropathy, and cancers[2,7], making NOX5 an attractive drug target[8,9]. Induced expression of NOX5 in transgenic mice accelerates renal injury in diabetic nephropathy[10], and NOX5 inhibitors manifest protective effects in that model[8].

NOX5 catalyzes the electron transfer from cytosolic NADPH to extracellular oxygen to produce superoxide ($O_2^-$) in the presence of intracellular $Ca^{2+}$ ions[3,11]. NOX5 shares a similar catalytic core with other NOX family members (NOX1-4 and DUOX1-2)[12] composed of a transmembrane domain (TMD) and a cytosolic dehydrogenase domain (DHD), which contains FAD-binding and NADPH-binding domains (FBD and NBD, respectively). Electron transfer occurs through the following path: NADPH − FAD − inner heme − outer heme − oxygen[13,14].

The enzymatic activity of NOX5 is activated by intracellular $Ca^{2+}$, but the underlying mechanism remains unclear. Like DUOX1-2, $Ca^{2+}$ activates NOX5 by binding to its regulatory EF-hand domain (EFD). However, activation mechanisms are not conserved between DUOX1-2 and NOX5 due to the different roles of EFD during activation[15,16]. Structures of cyanobacterial NOX5 (csNOX5) TMD and DHD[17], human NOX2-p22 complexes[18,19] and mouse and human DUOX1-DUOXA1 complexes[13,14] have been described. The structure of human DUOX1-DUOXA1 complex was determined both in the presence and absence of $Ca^{2+}$, revealing conformational changes in the EFD but almost identical in catalytic modules. The fundamental question regarding how electron transfer is initiated from NADPH to FAD in NOX5 or the rest of the NOX family remains unresolved. Biochemical analysis has also revealed two regulatory segments in NOX5: PhosR and REFBD[16]. Phosphorylation of the PhosR motif increases the $Ca^{2+}$ sensitivity of NOX5, while the REFBD motif is self-inhibitory. Yet, how these two segments modulate the function of NOX5 at the molecular level remains elusive.

[1]Department of Structural Biology, St Jude Children's Research Hospital, Memphis TN38105, USA. [2]Department of Chemical Biology & Therapeutics, St Jude Children's Research Hospital, Memphis TN38105, USA. [3]Division of Physics and Applied Physics, School of Physical and Mathematical Sciences, Nanyang Technological University, 673371 Singapore, Singapore. [4]These authors contributed equally: Chenxi Cui, Meiqin Jiang, Nikhil Jain. ✉e-mail: tanadet.pipatpolkai@ntu.edu.sg; ji.sun@stjude.org

In addition to $Ca^{2+}$, $Zn^{2+}$ and $H_2O_2$ could regulate the ROS production by NOX5 in sperm cells[20]. External application of $Zn^{2+}$, a critical message for sperm maturation and fertilization[21], can inhibit NOX5. The mode of action was proposed to be through inhibiting the Hv1 channel, which mediates the transport of protons generated during ROS production. However, it is unclear whether $Zn^{2+}$ can directly modulate the function of NOX5.

Here we determine cryo-EM structures of human dimeric NOX5 from pre- to post-reaction states, allowing us to dissect the molecular mechanism underlying $Ca^{2+}$-dependent NOX5 activation. Furthermore, we identified an unexpected putative cysteine-based $Zn^{2+}$-binding site, mutations of which disrupt the enzymatic activity of NOX5. Together with biochemical analyses, our study reveals the molecular basis of NOX5 functional modulation and ROS production.

## Results

### Structure characterization of the full-length human NOX5

To dissect the molecular basis of NADPH oxidase activation, we performed a structure-function analysis of NOX5, which contains a catalytic core and a minimal regulatory module compared to the rest of the NOX family members[11]. Human *NOX5* gene encodes six splicing isoforms, denoted as v1-v6 or α–ζ, with different tissue distributions[3–6]. Here we use the NOX5-β variant (hereinafter referred to as NOX5), which is functionally active and structurally compact. We overexpressed and purified the full-length NOX5 from HEK293 cells (Fig. 1a). To confirm and characterize the catalytic function of purified NOX5, we adapted the WST1-based activity assay[22]. The purified NOX5 showed a $K_{cat}$ of 30.33/min and a $K_m$ of 13.67 μM (Fig. 1b). To ensure the specificity of our enzymatic assay, we also tested NOX5

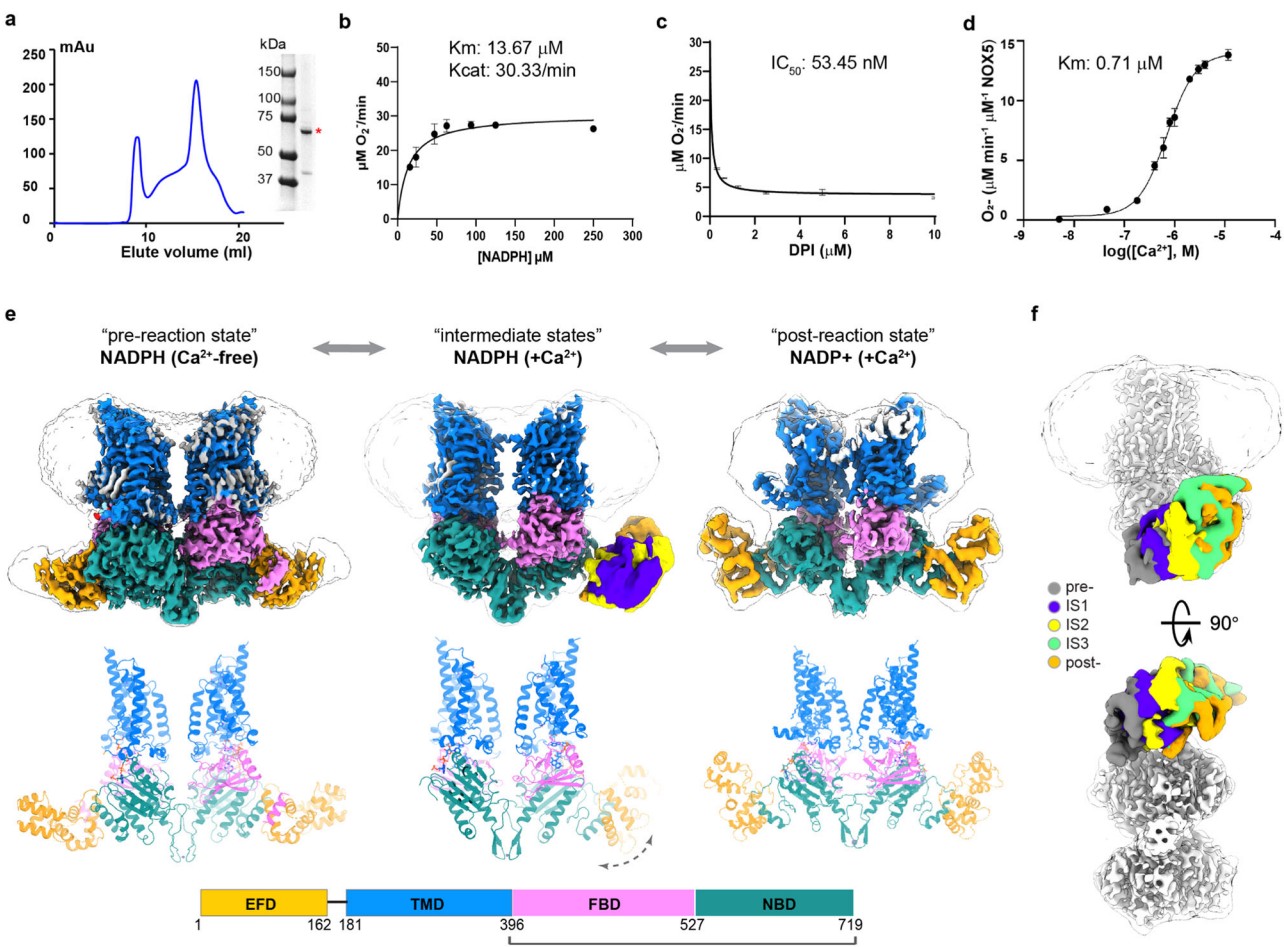

**Fig. 1 | Structural study of NOX5 in the pre-reaction, intermediate, and post-reaction states. a** Purification of human NOX5 in the $Ca^{2+}$-free condition using a Superose 6 increase column. SDS-PAGE shows the purified full-length human NOX5 with the peak fractions combined. This purification was repeated independently with similar results more than 3 times. Source data are provided as a Source Data file. **b** Enzymatic activity of NOX5 measured by WST-1 assay. Y-axis is the $O_2^-$ generation rate (μM/min), x-axis is the concentration of NADPH (μM). Data are shown (average +/-SD) with $n = 5$ biologically independent samples. Source data are provided as a Source Data file. **c** Activity inhibition of NOX5 by DPI. Y-axis is the $O_2^-$ generation rate (μM/min), x-axis is the concentration of DPI (μM). Data are shown (average +/-SD) with $n = 3$ biologically independent samples. Source data are provided as a Source Data file. **d** $Ca^{2+}$-dependent activation of NOX5. Y-axis is the $O_2^-$ generation rate (μM/min/μM (NOX5)), x-axis is the log value of $Ca^{2+}$ concentration (M). Data are shown as means ± standard deviations ($n = 5$). Source data are provided as a Source Data file. **e** Top: structures of NOX5 in the pre-reaction (EGTA +

NADPH), intermediate ($Ca^{2+}$ + NADPH) and post-reaction ($Ca^{2+}$ + NADP + ) states. Bottom: domain scheme of NOX5. EFD: EF-hand domain; TMD: transmembrane domain; FBD: FAD-binding domain; NBD: NADPH-binding domain; DHD: dehydrogenase domain. TMD, FBD and NBD are colored in blue, pink, and dark green, respectively. The same color scheme is used unless otherwise noted. In the intermediate states, EFD are colored in violet, yellow and orange, respectively. FBD and NBD form the dehydrogenase domain (DHD). **f** The different conformations of EFD in two views (side and bottom). Top and bottom panel represent side and bottom view, respectively. Here, the catalytic modules of different states from Fig. 1e are aligned to the consensus map of the intermediate state (white). Micelles are shown to indicate the membrane position, and only the consensus map of the intermediate state are shown for clarity. Different conformations of EFD are colored in dark gray (pre-reaction), violet (IS1), yellow (IS2), orange (IS3) and green (post-reaction). IS1-3 indicate intermediate state 1-3. All EFD are low pass filtered to 12 Å for comparison between different states.

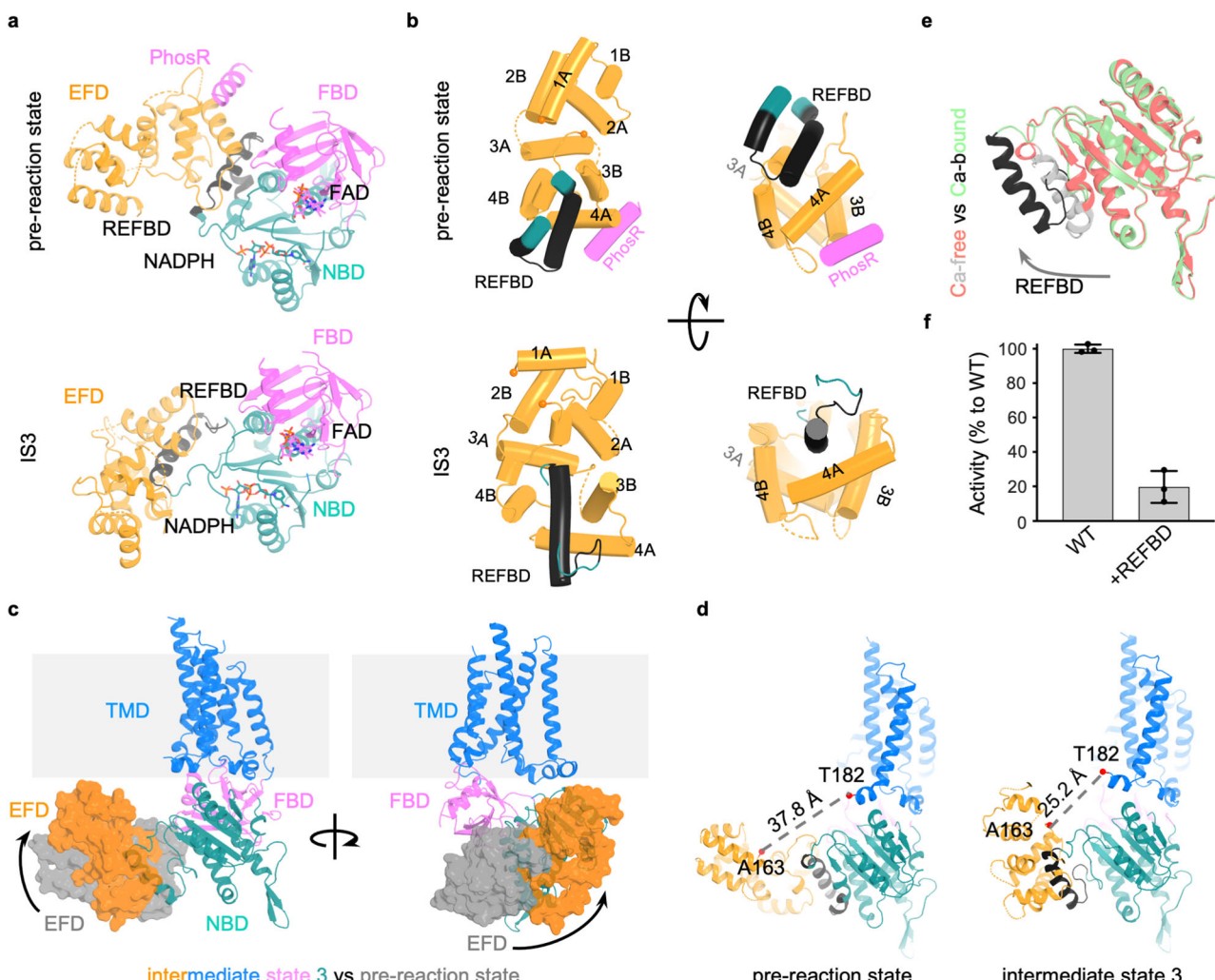

**Fig. 2 | Conformational changes of EFD. a** Conformational differences between pre-reaction state and intermediate state 3 in the cytosolic part of NOX5. **b** Conformational changes within EFD. The four EF hands are labeled with 1 A, 1B to 4 A, 4B. REFBD and PhosR segments are labeled. **c** The movement of EFD between intermediate state 3 and pre-reaction state. EFD color in orange and gray, respectively, in the intermediate state 3 and pre-reaction state. **d** Distance change between C-terminal of EFD and preTM1 in pre-reaction state and intermediate state 3. The REFBD motif is colored in black. **e** The movement of REFBD helix upon $Ca^{2+}$ binding. **f** REFBD inhibits the activity of NOX5. Y-axis is the relative activity compared with NOX5 wild type. x-axis indicates the NOX5 wild type and the NOX5 wild type incubated with REFBD. Data are shown (average +/-SD) with $n = 3$ biologically independent samples. Source data are provided as a Source Data file.

inhibition by diphenyleneiodonium (DPI) and calculated an $IC_{50}$ value of 53.45 nM (Fig. 1c). We then assessed $Ca^{2+}$-dependent activation, obtaining an $EC_{50}$ of 0.71 μM (Fig. 1d), similar to the previously reported value, 1.06 μM[11].

To understand the $Ca^{2+}$-dependent activation of NOX5, we determined cryo-EM structures of NOX5 in different catalytic stages: pre-reaction state (NADPH-bound without $Ca^{2+}$), intermediate states (in the presence of NADPH and $Ca^{2+}$), and post-reaction state (with NADP+ and $Ca^{2+}$) (Fig. 1e, Supplementary Figs. 1–3 and Supplementary Table 1) at overall resolutions of 3.2 Å, 3.3 Å and 4.1 Å, respectively, with C2 symmetry imposed. Focused refinement followed by symmetry expansion was performed to improve the local resolution of cytosolic domain of NOX5 in the pre-reaction state (see Methods).

### Overall structure of NOX5

Here we first use the pre-reaction state to describe the overall structure of NOX5. NOX5 forms homodimers rather than homotetramers as previously reported[23]. Each protomer contains a catalytic core with TMD, FBD and NBD and a regulatory EFD (Supplementary Fig. 4a, b). The catalytic module of NOX5 is similar to that of DUOX1-DUOXA1 complexes and csNOX5[13,14,17] (Supplementary Fig. 4c). Briefly, the TMD

contains six transmembrane helices (TM1–6) and binds two heme molecules (Supplementary Fig. 4d), and the cytosolic DHD binds FAD and NADPH (Fig. 2a). Similar to the csNOX5 and mouse DUOX1-DUOXA1 complex structures[14,17], we also observe lipid densities in close vicinity to the NADPH-binding site (Supplementary Fig. 4f, g), suggesting a conserved role for lipids in modulating the function of NOX proteins. Trp378 in csNOX5 was proposed to be important for electron transfer between the two hemes[17], but it is not conserved in human NOX5. Instead, the corresponding residue, Val362 and three other residues (Cys235, Val275 and Leu332) are located between the two hemes (Fig. 3a and Supplementary Fig. 4e) and might contribute to electron transfer. The distance between NADPH and FAD, which is >10 Å, is not optimal for transferring electrons (Fig. 3a and Supplementary Fig. 4e), as NADPH and FAD are solvent-exposed.

The catalytic module of NOX5 exhibits three insertions within the DHD, unique among NOX family members (Supplementary Figs. 4i and 5a). The first insertion (INS1, residues 480–515) is located within FBD and includes the previously identified PhosR segment (residues 489–505)[16], whose phosphorylation could boost the catalytic activity of NOX5. Part of INS1 displayed a helical density that should belong to PhosR, extending from the FBD and packed against the EFD

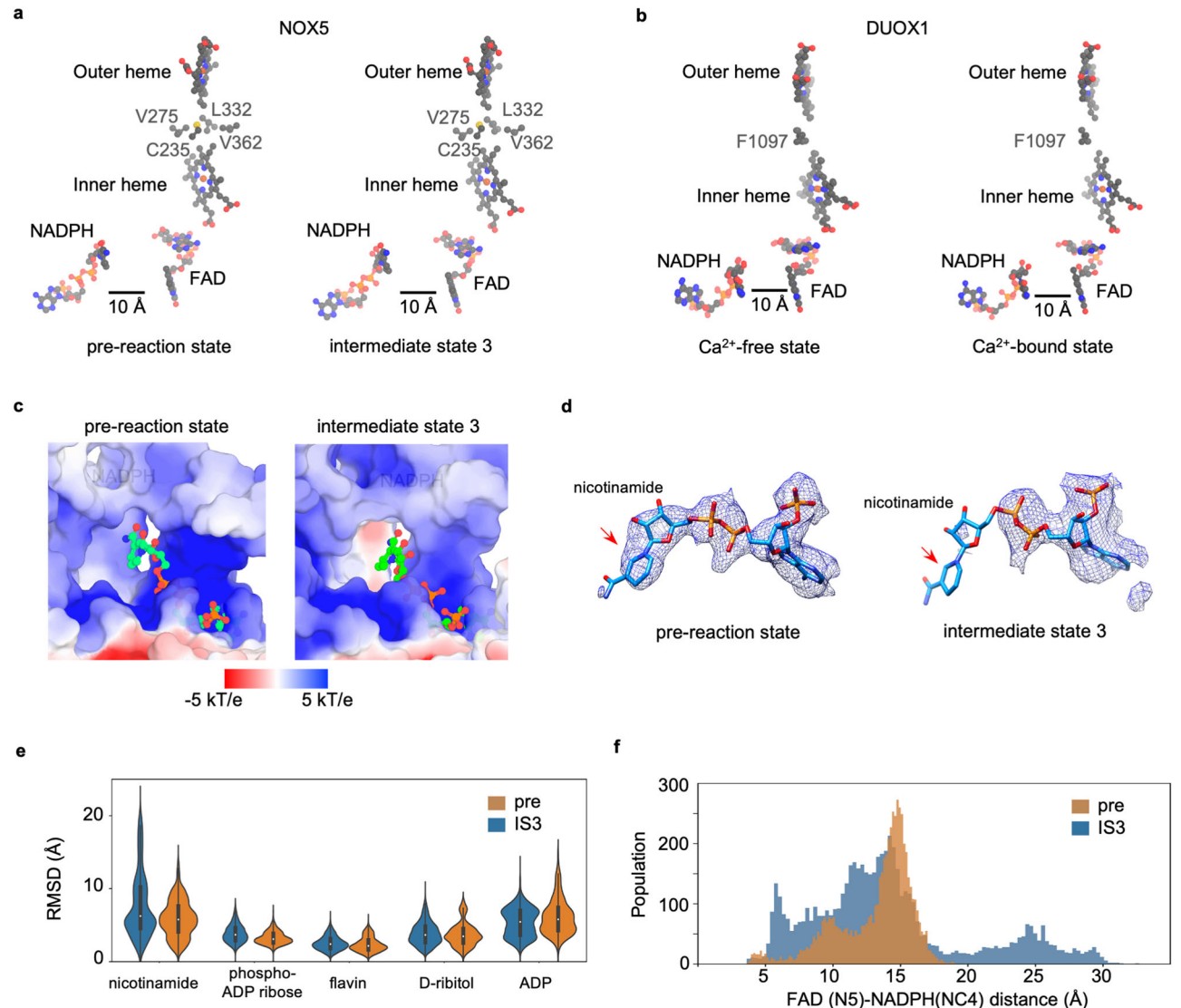

**Fig. 3 | Activation of human NOX5.** Electron transfer pathway in human NOX5 (**a**) and DUOX1 (**b**) before and after Ca²⁺ activation. Side chains of residues between two hemes are shown. **c** Electrostatic representation of the NOX5 NADPH binding pockets for pre-reaction state and intermediate state 3. **d** Cryo-EM density of NADPH in pre-reaction state and intermediate state 3. Nicotinamide group is indicated with red arrow. **e** The RMSD of chemical groups from NADPH and FAD in pre-reaction state and intermediate state 3. For pre-reaction state, the minima, maxima, centre, 25% percentile and 75% percentile of nicotinamide group are 0.03, 15.07, 5.77, 4.13 and 7.58, respectively. The minima, maxima, centre, 25% percentile and 75% percentile of phosphor-ADP ribose group are 0.03, 7.44, 3.10, 2.62 and 3.81, respectively. The minima, maxima, centre, 25% percentile and 75% percentile of flavin group are 0.04, 6.13, 2.17, 1.65 and 2.93, respectively. The minima, maxima, centre, 25% percentile and 75% percentile of D-ribitol group are 0.03, 8.96, 3.49, 2.60 and 4.49, respectively. The minima, maxima, centre, 25% percentile and 75% percentile of ADP group are 0.04, 15.86, 5.79, 4.32 and 7.41, respectively. For intermediate state 3, the minima, maxima, centre, 25% percentile and 75% percentile of nicotinamide group are 0.00, 22.51, 6.27, 4.58 and 10.20, respectively. The minima, maxima, centre, 25% percentile and 75% percentile of phosphor-ADP ribose group are 0.00, 8.23, 3.70, 2.95 and 4.62, respectively. The minima, maxima, centre, 25% percentile and 75% percentile of flavin group are 0.00, 6.54, 2.41, 1.86 and 3.11, respectively. The minima, maxima, centre, 25% percentile and 75% percentile of D-ribitol group are 0.00, 10.41, 3.66, 2.70 and 4.81, respectively. The minima, maxima, centre, 25% percentile and 75% percentile of ADP group are 0.00, 13.74, 5.47, 3.70 and 6.95, respectively. The bar covers 25% and 75% of the data. All data are plotted on the violin plot. Pre and IS3 represent pre-reaction state and intermediate state 3, respectively. Data in all triplicates are combined to a single histogram. **f** Distance distribution between FAD and NADPH in pre-reaction state and intermediate state 3. Pre and IS3 represent pre-reaction state and intermediate state 3, respectively. Data in all triplicates are combined to a single histogram.

(Fig. 2a, b and Supplementary Fig. 4j). The presence of this helix is supported by secondary structure prediction of PhosR (Supplementary Fig. 5b). The other two insertions (INS2, residues 564-578 and INS3, residues 647-654) are located in the NBD (Supplementary Figs. 4i and 5a). INS3 is part of the REFBD segment (residues 638-661) (Supplementary Fig. 5a), previously reported to play an inhibitory role on NOX5[16]. The TMD of NOX5 has short extracellular linkers (TM1-2, TM3-4 and TM5-6 linkers) (Supplementary Figs. 4d and 5a), resulting in

direct exposure of the oxygen binding site to the extracellular space (Supplementary Fig. 4e).

The EFD of NOX5 directly interacts with the DHD (Supplementary Fig. 4i). Sequence analysis suggests that NOX5 EFD contains four Ca²⁺-binding EF-hand motifs (EF1–4), three of which (EF2–4) have canonical sequences[24] (Supplementary Fig. 6a, b). In the pre-reaction state, EF1 and EF2 have a low local resolution (Supplementary Fig. 1a), which only allows the assignment of secondary structures (Supplementary

Figs. 1a and 6c). EF3 and EF4 are better resolved and interact with the DHD (Supplementary Fig. 6c, d). Specifically, the REFBD segment brings EF3-4 and the Rossmann fold of NBD together. Consistent with previous work[25] showing that two conserved Asp residues (D639 and D649) in *cs*NOX5 are important for DHD-EFD interaction, the corresponding two residues in the human NOX5 (D638 and D658) are located at REFBD (Supplementary Fig. 6h–i). The PhosR helix leans on the other side of EF3-4 (Fig. 2a, b and Supplementary Fig. 6e). This mode of interaction between EFD and DHD in NOX5 differs from that observed in human DUOX1 (Supplementary Fig. 6d, f), which might account for the different $Ca^{2+}$-dependent activation mechanisms in those two enzymes[14–16].

NOX5 dimerization is mediated by the cytosolic DHD through two interfaces (Supplementary Fig. 7a). The first interface involves unique residues in FBD and the linker connecting FBD and NBD (Supplementary Figs. 7a and 5a). Interface residues may include F422, H424, R426, R530, and R531, but their side chains are not well-resolved for detailed analysis. This interface is likely not as essential as the secondary interface for oligomerization, as point mutations (R426A, R530A or R531A) did not seem to disrupt NOX5 dimers (Supplementary Fig. 7c). The second interface involves residues from INS2 and likely forms a zinc-binding site (Supplementary Fig. 7a, b) which will be discussed later.

## $Ca^{2+}$-dependent conformational changes in EFD

In the consensus structure of intermediate state ($Ca^{2+}$- and NADPH-bound NOX5) determined at an overall resolution of 3.3 Å with C2 symmetry, the EFD was not resolved due to its flexibility (Supplementary Figs. 2a and 8a). To analyze the EFD motion, we performed 3D classification and refinement without applying symmetry (Fig. 1e, f and Supplementary Fig. 2a) and identified at least three distinct conformations (intermediate state 1–3) with large spatial displacement. The local resolutions of EFD in intermediate state 1 and 2 are insufficient for modeling secondary structure, whereas the intermediate state 3 was resolved to an overall resolution of 3.9 Å, enabling the model building of EFD (Supplementary Fig. 2a). These conformations, with the EFDs manifesting a trajectory towards the membrane, could represent its motions during NOX5 activation process (Fig. 1e, f and Supplementary Movie 1).

We further characterized the post-reaction structure of NOX5 in the presence of product (NADP + ) and $Ca^{2+}$ (Fig. 1e and Supplementary Fig. 3a). The EFD showed less flexibility than the intermediate states and is structurally similar to intermediate state 3 (Fig. 1e and Supplementary Fig. 8b), further supporting that the intermediate states 1–3 represent motions during NOX5 activation. For later analysis, we use the intermediate state 3 in structural comparison for its better resolution compared to the post-reaction state.

NOX5 EFD undergoes substantial conformational changes upon $Ca^{2+}$ binding (Fig. 2a–e), with EF1-2 moving toward the membrane (Figs. 1e, f and 2a, b). EF3-4 opens up, displaces the REFBD motif and detaches it from the NBD (Fig. 2c–e and Supplementary Fig. 6d, Supplementary Movie 1). REFBD was proposed to be a self-inhibitory peptide, and we confirmed that pre-incubation of purified NOX5 with synthetic REFBD peptides resulted in a significantly lower enzymatic activity (Fig. 2f). In addition, the PhosR region becomes flexible and not visible upon $Ca^{2+}$ binding (Fig. 2a, b).

To further explore the role of the EFD, we deleted it from NOX5 and generated the ΔEF construct (180-end), a construct that mimics the NOX5-ε splicing variant. As expected, the NOX5-ΔEF is no longer $Ca^{2+}$-sensitive (Supplementary Fig. 6g). However, to our surprise, the NOX5-ΔEF shows a low constitutive activity (Supplementary Fig. 6g), suggesting that EFD is important for inhibiting basal activity at low $Ca^{2+}$ conditions. Taken together, we conclude that the EFD shows a dual role in NOX5 activity: inhibitory at low $Ca^{2+}$ concentrations but activating at high $Ca^{2+}$ concentrations.

## Activation mechanism of human NOX5

We then explored the structure-function relationship between $Ca^{2+}$-induced conformational changes of EFD and NOX5 activation. First, we examined the electron transfer pathway. Surprisingly, the distance between NADPH and FAD, which is believed to be the speed-limiting step for electron transfer[26], is ~10 Å and not shortened upon $Ca^{2+}$ binding (Fig. 3a). The same observation was made in human DUOX1 (Fig. 3b). Previously, it was hypothesized that domain displacement between FBD and NBD might explain the $Ca^{2+}$-dependent activation[13], which was not observed in either human NOX5 or DUOX1 structures (Supplementary Fig. 9a).

Instead, we made the following structural observations. The EFD moves toward the membrane upon $Ca^{2+}$ activation, shortening the distance by more than 12 Å between EFD and TMD (Fig. 2d). The preTM1 helix, which caps the NADPH-binding pocket, uplifts (Supplementary Fig. 9b, c). The NBD shows a minor displacement coupled to the movement of the REFBD motif (Supplementary Fig. 9d). These conformational changes synergistically remodel the NADPH-binding pocket by enlarging the binding site and altering the local surface potential (Fig. 3c). Finally, the nicotinamide moiety of NADPH shows a weak density in the intermediate state 3 structure (Fig. 3d), suggesting the flexibility of this moiety as observed in the mouse DUOX1 structure[14].

The relaxation of substrate-binding pocket and weaker nicotinamide density (Fig. 3c, d) led us to hypothesize that the flexibility of the nicotinamide group could shorten the distance between NADPH and FAD and thus trigger electron transfer. To explore this possibility, we performed $3 \times 500$ ns all-atom molecular dynamics simulations[27] of the NADPH molecule in two protein conformations: the intermediate state 3, and the pre-reaction state. The all-atom simulations are required and sufficient for this study as we are interested in the local conformational changes of the co-factor within their binding pocket. The simulation results suggested that the nicotinamide part of NADPH indeed becomes more flexible in contrast to other parts of NADPH and FAD, as suggested by a larger deviation in RMSD (Fig. 3e) in the intermediate state 3 structure, consistent with the structural observation (Fig. 3c, d). As a result, the distance distribution between NADPH and FAD is much wider, and there is a higher probability that NADPH and FAD get close enough to initiate electron transfer (Fig. 3f). Consistent with the simulations, we see displacement of the NADP+ molecule in the post-reaction state relative to pre-reaction state (Supplementary Fig. 4f).

## A putative zinc-binding motif at the dimer interface

In both pre-reaction and intermediate state structures, we observe an ordered CXXC motif within the INS2 of NBD that contributes to NOX5 homodimerization (Fig. 4a and Supplementary Fig. 7a). INS2 is unique and conserved among NOX5 orthologs, and so is the CXXC motif (Fig. 4b), suggesting its functional importance. CXXC motifs are usually found in $Zn^{2+}$-binding and ROS-sensing proteins[28,29]. Our cryo-EM density and the orientation of the Cys residues led us to model a $Zn^{2+}$ between the two CXXC motifs at the dimer interface (Supplementary Fig. 7b). The presence of $Zn^{2+}$ was supported by the inductively coupled plasma mass spectrometry (ICP-MS) analysis[30], which showed enriched zinc ions in the NOX5 sample compared to buffer control (Fig. 4c). We then explored the role of the CXXC motif on NOX5 stability and activity by mutagenesis. Mutation of either cysteine (C568S or C571S) at the zinc finger could destabilize the NOX5 dimer (Supplementary Fig. 7d) and lead to a diminished enzymatic activity of NOX5 (Fig. 4d). Additionally, addition of TPEN, a zinc chelator, reduced NOX5 activity in a dose-dependent manner (Fig. 4e). These data suggest the CXXC motif at the dimer interface forms a zinc finger, which is important for NOX5 protein stability and kinase activity. Furthermore, it has been reported that zinc and redox signals can regulate NOX5-mediated ROS production in sperm[20]. Whether such regulation might also work directly through the cysteine-based zinc-binding motif we

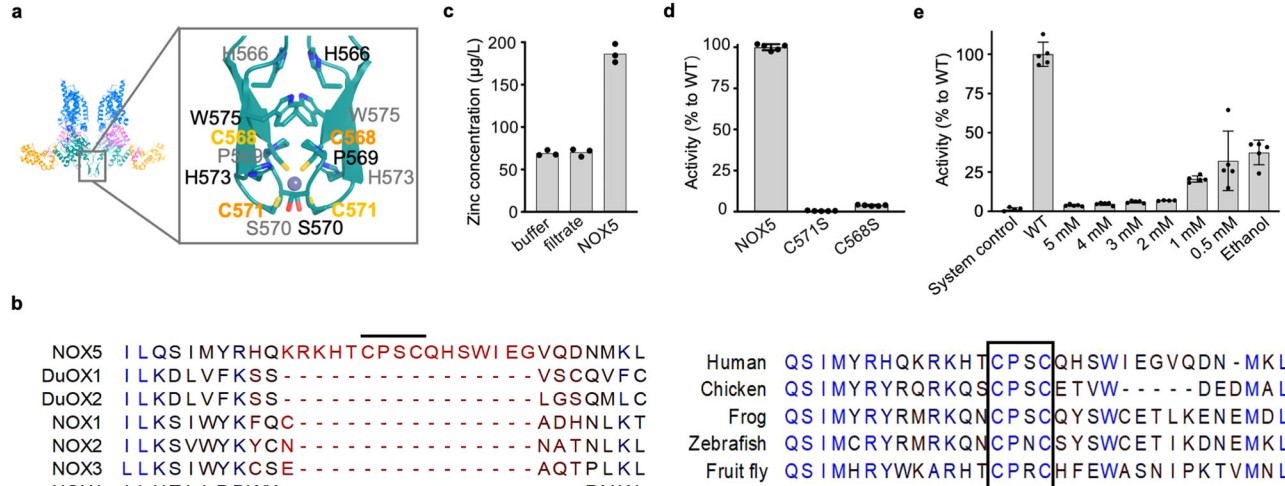

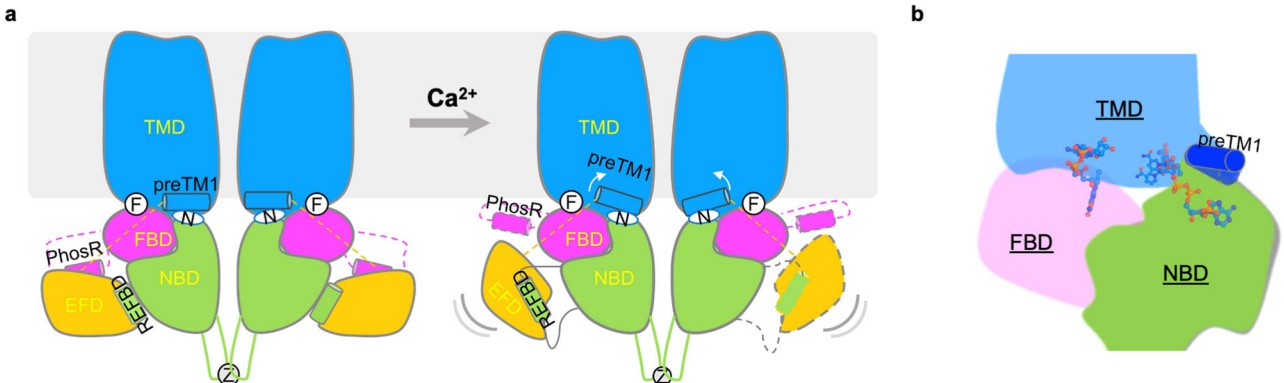

**Fig. 4 | The putative zinc binding site at the dimer interface. a** Structural details of the zinc binding site. **b** Sequence alignment of the CXXC-containing insertion between NOX5 paralogs and orthologs. The CXXC motif is indicated with black line and black box in left and right panel, respectively. **c** ICP-MS shows the presence of Zinc ions in NOX5. In x-axis: buffer indicates NOX5 sample buffer (20 mM Tris, 200 mM NaCl, 0.05% Digitonin); filtrate indicates the buffer passed through the filter device during NOX5 concentration; NOX5 is the experimental group. Data are shown with n = 3 technical repeats. Source data are provided as a Source Data file. **d** Relative activity of C571S and C568S to wild-type NOX5. Y-axis indicates the relative activity compared with NOX5 wild type. Data are shown (average +/-SD) with n = 5 biologically independent samples. Source data are provided as a Source Data file. **e** Dose-dependent inhibition of NOX5 by TPEN. Y-axis indicates the relative activity compared with NOX5 wild type. Data are shown with (average +/-SD) n = 5 biologically independent samples. Data are processed with GraphPad Prism (Version 10.1.1). We performed Grubbs' test[59] (also known as extreme studentized deviate, ESD), to determine the outliers. We excluded one data point in "2 mM TPEN" group, which is a significant outliner (P < 0.05) based on Grubbs' test. Source data are provided as a Source Data file.

**Fig. 5 | Working model of NOX5 activation. a** Conformational changes between pre-reaction and intermediate states of NOX5. The movement and flexibility of the EFD domain are illustrated. The upward rotation of preTM1 is indicated by white arrows. F in the white cycle stands for FAD, N in the white oval stands for NADPH, and Z stands for zinc. **b** The active site of NOX5 with flexible NADPH triggers electron transfer upon activation.

identified here is worth future investigation, for zinc fingers usually has picomolar affinity against zinc ions and maintains structural integrity of the protein.

## Discussion

In this study, we determined the cryo-EM structures of human NOX5 in the presence and absence of Ca²⁺ and captured the activation process in motion. Combining the structural observations and MD simulation data, we proposed a working model for NOX5 activation (Fig. 5). Briefly, upon Ca²⁺ binding, the EFD displaces the REFBD motif and slightly perturbs the NBD. At the same time, the upward movement of the EFD loosens the preTM1 helix, which sits on top of the NADPH binding site (Fig. 5a). These two molecular events together relax the NADPH-binding pocket, which leads to the higher flexibility of nicotinamide group of NADPH and triggers the electron transfer between NADPH and FAD and downstream superoxide production. Our model differs from the one proposed based on human DUOX1 structures[13], which suggested that domain displacement between NBD and FBD

would shorten the distance between NADPH and FAD and facilitate enzyme activation. As changes in distance between NBD and FBD are minimal in both human DUOX1 and NOX5 cryo-EM structures (Supplementary Fig. 9a), we think the domain displacement mechanism is unlikely.

The EFD plays a dual role in the activity of NOX5: inhibitory at low Ca²⁺ concentrations and activating at high Ca²⁺ concentrations (Supplementary Fig. 6g). We propose that the inhibitory role of EFD occurs through fixing the preTM1 and probably also the REFBD helix in the inactive state; removing the EFD unleashes the preTM1 and partially relaxes the NADPH-binding site, which in turn leads to low constitutive basal activity.

Our work also offers a possible explanation for the modulatory role of the PhosR segment, whose phosphorylation increases the Ca²⁺ sensitivity of NOX5[31]. We observed a helix from the PhosR motif in the pre-reaction structure but not intermediate state 3 structure (Fig. 2a, b), docking against a negatively charged surface on the EFD (Supplementary Fig. 10). PhosR may stabilize the EFD in the inactive

state, but this motif is released upon phosphorylation, facilitating the transition of the EFD to active conformations.

The CXXC motif at the dimer interface might be subject to various regulatory mechanisms. First, cysteines are sensitive to redox signals. Second, $Zn^{2+}$ concentration could have an impact on the structural integrity of the dimer interface. Third, the $Zn^{2+}$-binding region provides an extra surface for potential binding protein partners. Further investigations are needed to test these possibilities.

Overall, we unveiled the structural basis of $Ca^{2+}$-dependent activation of NOX5, which is subject to various regulatory signals, including $Ca^{2+}$, $Zn^{2+}$ and phosphorylation. Our work provides a structural framework for mechanistically investigating this crucial enzyme and developing therapeutic interventions.

## Methods

### Cell lines
Sf9 (Thermo Fisher, 12659017) cells were cultured in Sf-900 III SFM medium (GIBCO) at 27 °C. HEK293F (Thermo Fisher, R79007) cells were cultured in Freestyle 293 medium (GIBCO) supplemented with 2% FBS at 37 °C.

### Cloning, expression, and protein purification of NOX5
NOX5 v2 isoform was cloned with an N-terminal GFP tag in BacMan vectors using the Bac-to-Bac system according to the instruction in the manufacturer's manual (Invitrogen). All the mutations and deletion constructs were subcloned using the In-fusion cloning kit (Takara biosciences). For protein expression, 100 mL P2 virus was used for transfection to 1 L HEK293F cells grown in 293 freestyle media supplemented with 2% FBS. After 12 h, protein expression was induced by adding 10 mM sodium butyrate and cells were transferred to 30 °C and incubated for another 48−60 h[32]. Cells were harvested, and the pellet was resuspended in lysis buffer (20 mM Tris pH 8.0, 200 mM NaCl, 5 mM EGTA, 1 μM pepstatin, 2 μM leupeptin, 1 μg/mL aprotinin, and 100 μM PMSF). GDN or DDM/CHS (n-dodecyl-β-D-maltoside/cholesteryl hemisuccinate Tris salt, 5:1 mass ratio) was added to 1% concentration and nutated for one hour at 4 °C. Insoluble fractions were discarded after centrifuging at ~39,000 g for one hour. The supernatant was incubated with high-affinity 3 mL CNBR-activated sepharose beads (GE Healthcare) coated with GFP nanobodies (GFP-NB)[33] for 1.5 h at 4 °C. Incubated beads were passed through a gravity column, and beads were washed with 10 CV 0.02% GDN or DDM/CHS and 5 CV 0.05% Digitonin buffer prepared in 20 mM Tris pH 8.0, 200 mM NaCl and 5 mM EGTA. 1 mM adenosine triphosphate (ATP), 10 mM MgCl2 and 0.05% Digitonin buffer prepared in 20 mM Tris pH 8.0, 200 mM NaCl and 5 mM EGTA was used to remove heat shock proteins' contamination. GFP tag was cleaved by incubating beads at 4 °C with preScission protease for 12−16 h[34]. Flowthrough protein was concentrated and purified further by size exclusion chromatography using a Superose 6 column (Cytiva) with SEC buffer: 0.05% Digitonin in 20 mM Tris (pH 8.0), 200 mM NaCl and 5 mM EGTA. Different states of NOX5 samples were prepared through adding different reagents, which would be described below.

Peak fractions were concentrated, and aliquots were stored at −80 °C after snap freezing in liquid nitrogen for enzymatic assays. Protein concentration was determined by measuring the Heme amount using QuantiChrom™ Heme Assay Kit (Bioassay Systems).

### Trp fluorescence-based gel filtration chromatography
NOX5 mutations (R426A, R530A, R531A, R530A/R531A, C568S, C571S and C568S/C571S) were purified with the same protocol as NOX5 wild type. The GFP tag was cleaved by incubating beads at 4 °C with preScission protease for 2 hrs. Flowthrough protein was concentrated and centrifuged for 15 min with 20,000 g to remove aggregations. The supernatant was applied to a Supersoe 6 Increase column using the 20 mM Tris pH 8.0, 200 mM NaCl and 0.004% LMNG/CHS (8:1 mass

ratio) buffer as mobile phase. Tryptophan fluorescence was measured at the excitation wavelength of 280 nm and emission wavelength 350 nm.

### NADPH oxidase activity assay
NOX5 activity was measured using WST1 based assay[22] at room temperature. In brief, 0.27 pmols of NOX5 protein or mutant was added in 1X PBS containing 10 μM CaCl2, 3 μM EGTA, 1 mM MgCl2, 0.5 μM FAD, 0.2 mM NADPH and 0.2 mM WST1. Absorbance was measured at 438 nm every 30−60 s on 96-well plates in POLARstar Omega (BMG Labtech) plate reader. The amount of superoxide was calculated indirectly by converting OD to the amount of Formazan produced using its extinction coefficient ($37,000\ M^{-1}cm^{-1}$). The $Ca^{2+}$-dependent activity of NOX5 was performed in 100 μL of 1X PBS containing 0.5 μM FAD, 0.2 mM NADPH, 0.2 mM WST1 and 50 nM NOX5 in the presence of 0−11.7 μM free CaCl2 (Calcium Calibration Buffer Kit #1, Thermo Fisher). Concentrations of free $Ca^{2+}$ were determined by using fluorescent indicators Fura-2 (Thermo Fisher). $O_2^-$ generating reactions of NOX5 were monitored continuously by measuring absorbance at 438 nm every 55 s. To determine the $K_{cat}$ and $K_m$ values, we measured NADPH oxidase activity in the presence of varying concentrations of NADPH (0−250 μM). All data were analyzed by GraphPad Prism and Microsoft Excel. For determining Kcat and Km, Michaelis-Menton Kinetics ($Y = Vmax*X/(Km + X)$) (1) was used for curve fitting. The dose-response inhibition by DPI was plotted using the equation: $Y=Bottom + (Top-Bottom)/(1 + (X/IC50))$ (2) in GraphPad Prism. The $Ca^{2+}$-dependent activation was analyzed using the dose-response equation: $Y=Bottom + (Top-Bottom)/(1 + 10^{((LogEC50-X)*HillSlope)})$ (3).

NOX5 activity assay with TPEN was also measured using WST1 based assay[22] at room temperature. Briefly, 0.67 pmols of NOX5 protein was mixed with 10 μM CaCl2, then incubated with TPEN at final concentrations of 5 mM, 4 mM, 3 mM, 2 mM, 1 mM and 0.5 mM, respectively, for 10 mins. Here TPEN are dissolved in 100% ethanol, and the final ethanol concentration in the NOX5-TPEN mixture is ~8%. 1X DPBS containing 0.02% Digitonin, 100 μM EGTA, 0.5 μM FAD, 0.2 mM NADPH and 0.2 mM WST1 was then added. Absorbance was measured at 438 nm every 60 s on 96-well plates in POLARstar Omega (BMG Labtech) plate reader. The value of wild-type NOX5 activity without TPEN was 100%, other values with TPEN are relatively compared with wild-type NOX5.

### ICP-MS measurement
Sample preparation: NOX5 was purified with TBS buffer (20 mM Tris pH 8.0, 200 mM NaCl and 0.05% Digitonin), and concentrated to ~2 mg/mL. Filtrate was the buffer passed through the filter device during NOX5 concentration. NOX5 sample, TBS buffer and filtrate were measured ($n = 1$ biologically independent samples) with 3 technical repeats, respectively. TBS buffer and filtrate were measured as controls.

Digestion procedure: 0.5 mL of provided samples were placed in the Teflon vessels, and then 1 mL of concentrated Nitric Acid was added. The tubes were sealed. Then samples were placed into Mars 6 Microwave Digestion System (CEM Corporation). Microwave power was ramped up to 100 °C over 20 min, and then held at that temperature for 2 min. This 1.5 mL mixture was diluted by 28.5 mL deionized water to a final dilution factor of 60x before the ICP-MS analysis.

ICP-MS analysis: ICP-MS[35] was conducted by Chemical and Environmental Analysis Facility at Washington University in St. Louis with PerkinElmer NexION 2000 instrument. Software Synthetic (Build 2.4.8034.0) was used to process and analyze the measurement. The estimated total reading time for each sample was 5.02 s. The number of reading was 50, which was the total number of mass-to-charge ratios scanned during the analysis. The dwell time per amount of mass was 50 ms and the integration time was 2500 ms. Sample was read with scan mode of peak hopping. Isotopes of Zinc 64, 66, 67, 68 and 70

were all collected, but Zinc (65.926) was the mass to charge. Scandium (Sc) with mass amount 44.9559 was the element we selected as internal standard. Internal standard was measured as a quality control at the same time as samples' measurement. The stability and performance of the instrument over time was assessed by monitoring the response of the internal standard throughout the analysis. Internal standard was separate from samples. For sampling, sample was flushed with 60 s and speed of −75 rpm, read with delay of 30 s and −20 rpm, washed with 90 s and −75 rpm, and analyzed with −20 rpm. Standard samples of Zinc at concentrations of 1, 10, 50, 100 and 200 microgram per liter (ppb) were measured and a calibration curve (weighted linear) was created. The concentrations of Zinc in samples were calculated through the standard curve. Each standard and sample was measured triplicates, all the final concentration reading was the average of the triplicates reading.

## Molecular Dynamics (MD) Simulations

Short unresolved loops (between residues I20-I27, V38-E40, D51-T60, H73-S75, I89-D96, D132-N136 exclusive) within the structure of hNOX5 were modelled based on the sequence obtained from the Uniprot database (Q96PH1-4). Longer unresolved loops were modeled by trimming the loop to a shorter length to the following sequence and then connected with the peptide bond (A161-Q180: AHWLTAPAPR, Q295-H318: AEASPGIGWV and S479-C515: DPLEKLKF). All loops and missing side chains were modeled using SwissModel to prevent alteration in the solved structure[36]. The structure of either pre-reaction state or intermediate state 3 was embedded into the phosphatidyl oleyl phosphatidylcholine (POPC) bilayer and solvated in 0.15 M NaCl using CHARMM-GUI Membrane Builder[37–39]. Heme-B[40], NADPH parameters were obtained from CHARMM36m based on previous parameterization, whereas the parameters for $FADH_2$ were obtained from the default Ligand reader and modeler from CHARMM-GUI[41]. The production run were simulated with a 2 fs timestep using the CHARMM36m[42] forcefield in GROMACS2022.1[43]. The system was then energy minimized and equilibrated using the standard six steps CHARMM-GUI equilibration protocol. This includes the following set-up: The protein backbone was restrained at the force constant of 4000, 2000, 1000, 500, 200 and 50 $kJmol^{-1}nm^{-2}$, the protein side chain was restrained the force constant of 2000, 1000, 500, 200, 50, and 0 $kJ\,mol^{-1}nm^{-2}$, the lipids non-H atoms were restrained at the force constant of 1000, 400, 400, 200, 40 and 0 $kJ\,mol^{-1}nm^{-2}$ and the dihedral restraint was set at the force constant of 1000, 400, 200, 200, 100 and 0 $kJ\,mol^{-1}rad^{-2}$. The simulations were equilibrated with a 1 fs timestep for 125 ps for the first three steps, and then to a 2 fs timesteps for 500 ps the next two, and 5 ns for the final step. The first two steps were conducted with the NVT ensemble where the last four were conducted with the NPT ensemble. All equilibration runs were conducted at 310 K using Berendsen thermostat. In all NPT ensemble equilibration, the semi-isotropic pressure was maintained at 1 bar using Berendsen barostat. The production runs were conducted for 500 ns under 310 K using the v-rescale thermostat[44]. The semi-isotropic pressure of all systems was maintained at 1 bar using a C-rescale barostat[45]. All simulations were carried out in triplicates where velocity is randomly generated at the beginning of every simulation. All simulation frames were saved every 0.1 ns. All data from one condition is combined to a single histogram. All input files for production run (tpr), initial and the final co-ordinate files after 500 ns are available on https://doi.org/10.5281/zenodo.10947443.

## Cryo-EM sample preparation and data process

For pre-reaction state cryo-EM sample preparation, purified NOX5 (~8 mg/mL by OD280) was supplemented with EGTA and NADPH to final concentration of 5 mM and 1 mM, respectively. For the intermediate state sample, purified NOX5 was supplemented with $CaCl_2$ and NADPH to final concentration of 1 mM. Cryo-EM grids were prepared by adding

3.5 µL sample to the Quantifoil R1.2/1.3 holey carbon gold grids using a Vitrobot Mark IV system (FEI). Freezing was performed with a blot force of -3, a blot time of 3 s, and a wait time of 15 s under 100% humidity. NOX5 post-reaction state sample was prepared with EM GP2 (Lecia). Purified NOX5 (~10 mg/mL by OD280) was supplemented with $CaCl_2$, NADP+ and FAD to final concentration of 1 mM, 1.2 mM and 60 µM, respectively. The mixture was incubated for half an hour before freezing grids. 3.5 µL sample was applied and frozen with a blot time of 4 s, and a wait time of 5 s under 95% humidity at 8 °C.

Cryo-EM images were acquired by a 300-keV Titan Krios electron microscope (FEI) fitted with a K3 direct electron detector (Gatan) and an energy filter bioquantum. Images of pre-reaction and intermediate states datasets were recorded using SerialEM (version 3.7)[46] with a physical pixel size of 0.826 Å and a defocus range of −0.8 to −1.6 µm. Data were collected at a dose rate of ~25 e/Å²·s, and images were recorded during a 2.4-s exposure with 40-ms subframes (60 total frames). Super-resolution image stacks were gain-normalized[47], binned by 2 with Fourier cropping, and corrected for beam-induced motion using MotionCor2[48]. The data process was carried out in CryoSPARC (version 3.3)[49] and RELION4[50,51] (Supplementary Figs. 1–3), similar to the mouse DUOX1 work[14].

We collected two datasets for post-reaction state sample adopting EPU2 software with the same pixel size of 0.6485 Å. First dataset was collected with a defocus range of −1.0 to −2.6 µm and dose rate of 16.238 e/Å²·s. The total dose is 76.886 e/Å² (total frame number: 70). The second dataset was collected with a defocus value range of −1.0 to −2.6 µm and dose rate of 15 e/Å²·s. The total dose is 52 e/Å² (total frame number: 50). Data process was also carried out in CryoSPARC (Supplementary Fig. 3a).

For pre-reaction state, symmetry expansion followed by focused refinement was performed to improve the local resolution of cytosolic domains. Mask surrounding DHD and EFD was generated as follows: refined map was imported into ChimeraX[52] using volume eraser to remove transmembrane domain, cytosolic domains (DHD and EFD) map (map 1) was saved. Command line (relion_image_handler with the options --lowpass 15 --angpix 0.826) was used to create a low-pass filtered map for cytosolic domains, the low-pass filtered map was opened in ChimeraX to test the threshold (value 1) at which map showed no noisy spots outside the protein area. Then cytosolic domains map (map 1) was input into Mask Creation of RELION[50,51] with the options: Lowpass filter map (A) of 15, Initial binarization threshold of value 1, Extend binary map this many pixels and Add a soft-edge of this many pixels of 6 to generate the mask of cytosolic domains (DHD and EFD). For intermediate states, a heterogenous refinement followed by non-uniform refinement was performed to classify out different states without applying C2 symmetry. Local resolution was estimated using half maps as inputs in ResMap software[53]. Model validation was done in Phenix[54].

## Model building and refinement

Models were built in Coot[55]. First, homology models of NOX5 were generated based on *cs*NOX5 structures (PDB 5O0T and 5O0X)[17] and docked into the cryo-EM map. From this starting point, manual rebuilding was carried out. The structural model was refined using phenix.real_space_refine with secondary structure restraints and Coot iteratively. Protein structure quality was monitored using the Molprobity server[56]. Figures were prepared using PyMOL (The PyMOL Molecular Graphics System, Version 2.4 Schrödinger, LLC.) and UCSF Chimera[57]. The electrostatic representation in all figures was calculated using PyMOL-APBS Plugin[58]. Calculations were performed at 0.15 M ionic strength in monovalent salt, 298.15 K, protein dielectric 2, and solvent dielectric 78.

## Reporting summary

Further information on research design is available in the Nature Portfolio Reporting Summary linked to this article.

## Data availability

The cryo-EM maps of the NOX5 in pre-reaction, intermediate and post-reaction states have been deposited in the Electron Microscopy Data Bank under the accession codes EMD-42014 (pre-reaction), EMD-42015 (intermediate, consensus), EMD-42016 (IS3) and EMD-42013 (post-reaction). The corresponding coordinates have been deposited in the Protein Data Bank under the accession codes 8U85, 8U86, 8U87 and 8U7Y. NOX5 coordinates are generated based on homology models of *cs*NOX5, corresponding accession codes are 5O0T and 5O0X. For MD simulation, all input files for production run, initial and the final co-ordinate files after 500 ns are available on https://doi.org/10.5281/zenodo.10947443. Source data are provided with this paper.

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

## Acknowledgements

We thank the all members of the Cryo-electron Microscopy and Tomography Center of St. Jude Children's Research Hospital for help with cryo-EM data collection, Ines Chen and Zhaowen Luo for suggestions and help with bio-illustration during manuscript preparation, Junmin Peng and Boer Xie for help with mass-spectrometry analyses during the study. We thank all members of the Delemotte lab and the Sun lab for the fruitful discussion. The computations were enabled by resources provided by the Swedish National Infrastructure for Computing (SNIC) at the PDC Center for High Performance Computing, KTH Royal Institute of Technology, partially funded by the Swedish Research Council through grant agreement no. 2018-05973. This work was funded by the NIH (R01GM141357) and American Lebanese Syrian Associated Charities (ALSAC).

## Author contributions

C.C., M.J., N.J. and J.S. designed the project. C.C., M.J., N.J. and J.S. performed sample preparation and biochemical and structural analysis. Y.L. and A.A.K. helped with cryo-EM data collection. S.D and T.P. performed and analyzed the MD simulation with input from C.C. and J.S. C.C., M.J., T.P. and J.S. wrote the manuscript with input from all authors.

## Competing interests

The authors declare no competing interests.
