## [Peer Review File · Nature Communications]

REVIEWER COMMENTS

Reviewer #1 (Remarks to the Author):

Structural basis of Human NOX5 activation

Cui et al

Key results

In Cui et al., the authors present cryo-EM-based models of NADPH oxidase 5 (NOX5), an oxidative signaling enzyme that produces superoxide in response to calcium binding to its EF-hand domains. Superoxide is produced by the electron transfer from NADPH to FAD through two heme groups and finally to molecular oxygen. NOX5 regulates various physiological processes, and upregulation of NOX5 has been associated with human diseases; therefore, understanding details of NOX5 structure-function could contribute to the development of therapeutics. The authors determined cryo-EM structures of NOX5 in the pre-reaction state (NADPH-bound, no calcium), intermediate states (NADPH and calcium), and post-reaction state (NADP⁺ and calcium). In all states, NOX5 was a homodimer with similar conformations aside from the EF-hand domain. Overall, the model produced from the cryo-EM data fits well with previously published structures and models of NOX5 and the related family member DUOX1. The most novel findings of this manuscript are: 1) that calcium binding to the EF-hand domain results in increased dynamics of the dehydrogenase domain, specifically the NADPH binding site. These increased dynamics would allow the NADPH and FAD co-factors to sample positions close enough for electron transfer to occur. This answers a question in the field regarding how this electron transfer could occur since in NOX5 and DUOX1 structures, the NADPH and FAD are observed too far apart for electron transfer. It also suggests a novel regulatory role of NOX5 dynamics in the initiating step of electron transfer. 2) The authors identified a possible zinc-binding site near the dimer interface, which may play a role in oligomerization and activity.

The authors have revealed novel insights into the activation of NOX5 through structural dynamics, which will be of interest to the field of oxidative signaling and could have implications for future translational work. The techniques are sound and, aside from minor issues, are well-described in the methods. Most of the conclusions are supported by the data. The manuscript is well-written and adds to our understanding of NOX5 structure and function. However, there are some concerns that should be addressed, most significantly involving the proposed zinc-binding site and discussion of how this new model of NOX5 structure fits with previous reports regarding its oligomerization state, stabilization of the DH domain, and EF-hand dynamics. These concerns are outlined in more detail below.

Major concerns

As the discovery of a possible zinc-binding site is one of two major findings of this paper, it needs to be

explored further. The authors did not see any density for a bound zinc, but were able to model one in. They also found that mutation of the Cys residues resulted in a disruption of the homodimer and inactivation of NOX5. However, the authors did not show that zinc is actually required for dimerization or activity, as the result could be due to the loss of disulfide bonds, or other disruptions to the structure (and therefore activity). Activity and oligomerization assays with zinc chelation and zinc concentration series could support their claim that there is a zinc-binding site and that it is directly involved in NOX5 dimerization and activation.

Further, as the cited literature suggests that zinc inhibits NOX5 activity, how do the authors fit this published data with the proposal that the zinc-binding site is required for oligomerization and activity (vs inhibitory either directly or through the Hv1 channel)?

A wider discussion of the NOX5 literature, in terms of structure-function and regulatory mechanisms is needed. Some examples include;

Kawahara et al 2011 showed that oligomerization occurred through the DH domain and that the functional oligomer may be a tetramer (ref 1, see below)

Ref 15 proposes an oxygen binding site, how does this fit with the cryo-EM model presented here?

DH domain interactions with Hsp90 have been shown to be a crucial regulatory mechanism. How does this fit with the EFD DH domain dynamics, oligomerization and proposed conformational changes? (refs 2-4, see below)

How does the structural flexibility and dynamics of the EF domain compare to that seen in Fananas et al 2019, in which they see the csNOX5 EFs are partially unfolded in the absence of Ca, and identify conserved aspartates that may be important for DH-EF domain interactions? (ref 5, see below)

Other concerns

As oligomerization is thought to be required for activity, details of the dimer interface will be of interest to the field. Therefore, details of the dimer interface should be explained further. The first interface is only described in one sentence (line 124), and the corresponding figure, S5a doesn't make it easy to determine the protomer:protomer interactions. For example, are the Arg residues at the interface? If so, which amino acids are they interacting with? If this is unknown because the FBD:NBD linker is unstructured (and therefore invisible to the cryo-EM), this should be explained. A brief discussion and a figure in which the two protomers are shown in different colors, and the potential interacting amino acids highlighted would be helpful. This goes for site #2 as well, by coloring the protomers in different colors, the readers would be able to easily see the interface.

Ideally, if this the real dimer interface, mutation of the residues (e.g. Arg) would clearly disrupt it.

Abstract, line 15, there is no need for the "The" before NADPH oxidase 5.

Intro line 48, this should be re-worded to be more precise. As stated in line 46, Ca²⁺ binding activates DUOX1/2 and NOX5, the difference, as determined by the 2 references stated, is that Ca binding to DUOX1 relieves an EFD-DH autoinhibitory interaction while Ca binding to NOX5 relieves a DH-DH (REFBD) autoinhibitory interaction. Additionally, this does not necessarily fit with the extensive EFD-DH interactions seen in the cryo-EM model of DUOX1 in the presence of high Ca levels (ref 11).

All the figures should be larger.

More details are needed for Fig 1a. Is this the final SEC with Superose 6? Is this the Ca-bound or Ca-free prep? Which fractions were collected (and correlate to the SDS page inset)? What is the presumed oligomeric state at this step?

The equations used for the determination of Km, Kcat, and IC50 should be reported (either in results, methods, or figure legend).

Fig 1: the labeling in f should be made darker (it is so light, especially the yellow and orange, it is hard to read)

The green vs. teal is very hard to distinguish (Fig, 2e, and S4f)

Fig S3b, please state how the secondary structure predictions were determined (in text, figure legend, or methods)

The text in the results section and Figs 2 and 3 are not well aligned. After Figs 1, S1 and S2, Fig 2c is referenced on lines 94-95, then 3a on lines 100 and 102, 2c-d on line 108 and 120, Fig 4a on line 125 and finally 2a-e on line 143. It would be helpful for the figures to be in the order in which they are referenced in the text.

There does not seem to be a movie as referenced in the text.

The csNOX5 and DUOX1 structures appear to have a different orientation of the preTM vs. the NOX5 model (Fig 2f v 2g). Do the authors think this is significant?

“Grabs” is perhaps not the most precise word for the REFBD:EF interaction and associated conformational changes (lines 144 and 203)

WST1 assay for formazan production. Are these all from the same protein prep, or multiple protein preps?

For Fig. S4g, is this a representative trace or an average of multiple biological replicates?

Line 157: “We ask how conformational changes in EFD upon Ca²⁺ binding lead to NOX5 activation.” Could be re-phrased.

Line 181, “In” is not needed before “consistent with the simulation....”

Line 273; NOX5 conc was determined by measuring heme, but NOX5 is not always heme saturated in the cell 4 (unless excess heme was supplied during expression). Then in line 315, it is stated that conc of NOX5 was determined by OD280. So which assays/experiments used the quantification done with heme?

Line 288, extinction coefficient should have units

Were all activity assays conducted at room temp? it only specifically states for calcium conc series.

The font changes in lines 327-339

Abbreviations: FSC, FSEC, should be defined. Alternatively, since it is referred to simply as tryptophan fluorescence in Fig. S5c, the abbreviation FSEC may not be needed at all.

Introduction, lines 31-33 are missing references; Banfi 2001 showed NOX5 expression in testis, spleen and lymph nodes, Bedard, 2012 is a review citing expression of NOX5 in the following tissues; spleen, testis, placenta, uterus, ovary, lymph nodes, pancreas and cells; endothelial cells, VSMCs, cardiac fibroblasts.

NOX5 expression in oligodendrocytes (driving oligodendrocyte differentiation) was shown in 2016 by Acetta et al (ref 6, see below) NOX5 expression in cardiomyocytes was shown in 2012 by Hahn et al. (ref 7, see below) This is not a comprehensive list of tissues and cell types in which NOX5 has been identified, but does cover the tissues and cells listed by the authors.

Phenix needs a ref (line 341) Liebschner et al 2019.

Line 344, please add the pdb(s) and citation

References 11 and 24 are the same paper

1. Kawahara T, Jackson HM, Smith SM, Simpson PD, Lambeth JD. Nox5 forms a functional oligomer mediated by self-association of its dehydrogenase domain. *Biochemistry*. 2011;50(12):2013-25. Epub 2011/02/16. doi: 10.1021/bi1020088. PubMed PMID: 21319793; PMCID: 3073450.
2. Chen F, Pandey D, Chadli A, Catravas JD, Chen T, Fulton DJ. Hsp90 regulates NADPH oxidase activity and is necessary for superoxide but not hydrogen peroxide production. *Antioxidants & redox signaling*. 2011;14(11):2107-19. doi: 10.1089/ars.2010.3669. PubMed PMID: 21194376; PMCID: 3085945.
3. Chen F, Haigh S, Yu Y, Benson T, Wang Y, Li X, Dou H, Bagi Z, Verin AD, Stepp DW, Csanyi G, Chadli A, Weintraub NL, Smith SM, Fulton DJ. Nox5 stability and superoxide production is regulated by C-terminal binding of Hsp90 and CO-chaperones. *Free radical biology & medicine*. 2015;89:793-805. doi: 10.1016/j.freeradbiomed.2015.09.019. PubMed PMID: 26456056.
4. Sweeny EA, Schlanger S, Stuehr DJ. Dynamic regulation of NADPH oxidase 5 by intracellular heme levels and cellular chaperones. *Redox biology*. 2020;36. doi: 10.1016/j.redox.2020.101656.

5. Millana Fananas E, Todesca S, Sicorello A, Masino L, Pompach P, Magnani F, Pastore A, Mattevi A. On the mechanism of calcium-dependent activation of NADPH oxidase 5 (NOX5). *FEBS J.* 2020;287(12):2486-503. Epub 20191220. doi: 10.1111/febs.15160. PubMed PMID: 31785178; PMCID: PMC7317449.
6. Accetta R, Damiano S, Morano A, Mondola P, Paternò R, Avvedimento EV, Santillo M. Reactive Oxygen Species Derived from NOX3 and NOX5 Drive Differentiation of Human Oligodendrocytes. *Frontiers in Cellular Neuroscience.* 2016;10. doi: 10.3389/fncel.2016.00146.
7. Hahn NE, Meischl C, Kawahara T, Musters RJ, Verhoef VM, van der Velden J, Vonk AB, Paulus WJ, van Rossum AC, Niessen HW, Krijnen PA. NOX5 expression is increased in intramyocardial blood vessels and cardiomyocytes after acute myocardial infarction in humans. *The American journal of pathology.* 2012;180(6):2222-9. Epub 2012/04/17. doi: 10.1016/j.ajpath.2012.02.018. PubMed PMID: 22503554.

Reviewer #2 (Remarks to the Author):

The manuscript entitled 'Structural Basis of Human NOX5 Activation' by Cui and coworkers used cryo-EM, mutagenesis, and molecular dynamics (MD) simulations to characterize the structures of human NOX5 and propose a working model for NOX5 activation. The study presents valuable insights into the molecular mechanisms of NOX5 activation. Overall, the paper is well-structured and the figures are informative. However, there are several areas that require clarification and expansion for a comprehensive understanding.

The paper lacks sufficient technical details regarding the cryo-EM experiments, including describing the resolution of the different state models in the results section. Furthermore, how does the local resolution and features change when the models are computed without C2 symmetry? For the focus refinement analysis, how were the masks created? What is the local resolution of the ligands and their environments?

Details of MD simulations: the authors say that the structure of the system was minimized and equilibrated using 'standard CHARMM-GUI' protocol. More details should be added to describe the MD simulations because these protocols are not necessarily known by all potential readers and can be changed by the CHARMM-GUI developers at any time. Also, the authors do not provide any detailed analysis to assess if the simulations were well equilibrated after 500 ns. Furthermore, which fraction of these 500 ns was used, for example, for figure 3.e?

No details are given on how the electrostatic potentials in fig. 3 and S8 were obtained. Some methods can have serious pitfalls, so explicitly citing which approach was used would help the reader interpret these figures.

The authors postulate that the 'flexibility of the nicotinamide group could shorten the distance between NADPH and FAD and thus trigger electron transfer'. However, they do not provide a molecular rationale

in which the local environment of the nicotinamide group is explored in the structures and/or MD simulations. How do the coordination of the different groups change in the different states?

Fig 1.a: provide more detailed legend for the left panel.

Fig 1.e: indicate which domains encompass the DH.

Fig 1.f: this panel is unclear to me. Please indicate what the top and bottom panel represent.

Fig 2.b: please indicate which helix is shown in black.

Fig 2: It is difficult to visualize the statement 'EF3-4 opens up, grabs the REFBD motif and detaches it from the NBD'. In particular, it is not clear if there is a binding effect or if the REFBD conformational change is just a consequence of the connectivity of the protein. Is it really that the REFBD protein-protein interface is changing from EF3-4 to the NBD?

Fig 3: I would suggest using the same naming and order when referring to the different states. In this figure using Ca²⁺-free state, Ca²⁺-bound state, intermediate state-3, Ca²⁺, and EGTA is confusing about the equivalence between states.

In conclusion, the paper offers valuable insights into the molecular mechanisms of NOX5 activation and its structural characteristics. To enhance its clarity and impact, the paper should provide more straightforward explanations of the experimental and computational methods. Providing more technical details would strengthen the paper's overall quality.

Reviewer #3 (Remarks to the Author):

NADPH oxidase 5 (NOX5) of the NOX family catalyzes production of superoxide by transferring electrons from cytosolic NADPH to extracellular oxygen. Its enzymatic activity is regulated by multiple intracellular factors, such as Ca²⁺, Zn²⁺, and phosphorylation. Ca²⁺ is required for the activation of NOX5; however, its activation mechanism is unclear due to the lack of high-resolution structural information. This manuscript by Cui and coauthors reports the cryo-EM structures of NOX5 in three different states: pre-reaction, intermediate, and post-reaction states. By interpreting these structures and supported by mutagenesis analyses and MD simulations, the authors propose a mechanism for NOX5 activation by Ca²⁺ that the binding of Ca²⁺ in the EF-hand domain (EFD) leads to motions of the EFD and increases dynamics of NADPH, therefore allowing electron transfer from NADPH to FAD. This is an intriguing new discovery that will be an important step to advance our understanding of the mechanisms of activation/regulation and electron transfer in NOX5 and more broadly in the NOX family.

Specific comments:

1. The methods for cryo-EM sample preparation of NOX5 in the three different conditions are not clearly described. It is useful and important to provide sufficient details, such as the concentration of each key compound, incubation time, etc.
2. The authors did meticulous analyses of the cryo-EM data and found populations of particles in different conformations. However, they did not discuss possible variations caused by cryo-EM grid

preparation. There are a few illy controlled factors in cryo-EM grid preparation, such as interactions of protein with the air/water interface, ice thickness, increase of salt/detergent concentration caused by evaporation during grid blotting, etc. The reviewer suggests the authors repeat their cryo-EM experiments to confirm that conformational changes/motions are truly from Ca²⁺ binding and not other unrelated variations.

3. The 2D class averages in Fig. S1 show dimers that vary on the EFD. Some have two EFDs and others have only one. It is interesting that this difference is orientation dependent. The reviewer is concerned that one of two EFDs interacts with the air/water interface and becomes mobile or unfolded. It is also possible that the EFD is invisible at some angles when it is overlapping with the TMD. It is a concern that the authors obtained asymmetric structures of the intermediate states with only one EFD, although the 2D class averages show a good 2-fold symmetry and both EFDs in the dimer.

4. The cryo-EM density of the Ca²⁺ bound intermediate state 3 shows weak density of the nicotinamide moiety of NADPH. The authors suggest this is caused by an increased flexibility of this moiety. But this may also be a result of the lower resolution of this structure. Also, the consensus structure of the Ca²⁺ bound state seems to show a strong density of the nicotinamide moiety (Fig. 2f).

5. The authors should show densities of both NADPH and NOX5 in Fig. 3d for a more reliable interpretation of the cryo-EM map.

6. The reviewer is not an expert on MD simulations, but feel cautious about the simulations results in the paper because the models used in the simulations contain a few low-resolution domains and missing loops/side chains that are modelled using SwissModel.

7. The conclusion about the Zn-binding motif is speculative because neither a clear density of Zn²⁺ ion is resolved in the cryo-EM maps nor experiments were performed to confirm it is Zn²⁺ and not other ions. It is more precise to call it a putative Zn-binding motif.

POINT-BY-POINT RESPONSE TO REVIEWERS' COMMENTS

Reviewer #1

In Cui et al., the authors present cryo-EM-based models of NADPH oxidase 5 (NOX5), an oxidative signaling enzyme that produces superoxide in response to calcium binding to its EF-hand domains. Superoxide is produced by the electron transfer from NADPH to FAD through two heme groups and finally to molecular oxygen. NOX5 regulates various physiological processes, and upregulation of NOX5 has been associated with human diseases; therefore, understanding details of NOX5 structure-function could contribute to the development of therapeutics. The authors determined cryo-EM structures of NOX5 in the pre-reaction state (NADPH-bound, no calcium), intermediate states (NADPH and calcium), and post-reaction state (NADP⁺ and calcium). In all states, NOX5 was a homodimer with similar conformations aside from the EF-hand domain. Overall, the model produced from the cryo-EM data fits well with previously published structures and models of NOX5 and the related family member DUOX1. The most novel findings of this manuscript are: 1) that calcium binding to the EF-hand domain results in increased dynamics of the dehydrogenase domain, specifically the NADPH binding site. These increased dynamics would allow the NADPH and FAD co-factors to sample positions close enough for electron transfer to occur. This answers a question in the field regarding how this electron transfer could occur since in NOX5 and DUOX1 structures, the NADPH and FAD are observed too far apart for electron transfer. It also suggests a novel regulatory role of NOX5 dynamics in the initiating step of electron transfer. 2) The authors identified a possible zinc-binding site near the dimer interface, which may play a role in oligomerization and activity.

The authors have revealed novel insights into the activation of NOX5 through structural dynamics, which will be of interest to the field of oxidative signaling and could have implications for future translational work. The techniques are sound and, aside from minor issues, are well-described in the methods. Most of the conclusions are supported by the data. The manuscript is well-written and adds to our understanding of NOX5 structure and function. However, there are some concerns that should be addressed, most significantly involving the proposed zinc-binding site and discussion of how this new model of NOX5 structure fits with previous reports regarding its oligomerization state, stabilization of the DH domain, and EF-hand dynamics. These concerns are outlined in more detail below.

We sincerely appreciate the positive comments of the reviewer.

Major concerns:

As the discovery of a possible zinc-binding site is one of two major findings of this paper, it needs to be explored further. The authors did not see any density for a bound zinc, but were able to model one in. They also found that mutation of the Cys residues resulted in a disruption of the homodimer and inactivation of NOX5. However, the authors did not show that zinc is actually required for dimerization or activity, as the result could be due to the loss of disulfide bonds, or other disruptions to the structure (and therefore activity). Activity and oligomerization assays with zinc chelation and zinc concentration series could support their claim that there is a zinc-binding site and that it is directly involved in NOX5 dimerization and activation.

Thanks for the comments. We did observe potential zinc density in our cryo-EM map (Fig. S5b) and have followed the reviewer's suggestions and further explored the zinc-binding site in our revision:

- we detected zinc ions in purified NOX5 using inductively coupled plasma mass spectrometry (ICP-MS), an analytical technique that can measure elements at trace levels in biological samples (Fig 4c).

- we tested NOX5 activity in the presence of different concentrations of TPEN, a zinc chelator, and observed dose-dependent inhibition of NOX5 by TPEN (Fig. 4e).

These two pieces of evidence, combined with the cysteine mutagenesis data (cysteine mutation reduces NOX5 stability and activity and alters oligomerization state), support our interpretation of the cryo-EM density map that a zinc ion is likely present at the dimerization interface, and that it is important for the structure and the enzymatic activity of NOX5. We have thus revised our manuscript accordingly (lines 197-205).

“The presence of Zn²⁺ was supported by the inductively coupled plasma mass spectrometry (ICP-MS) analysis, which showed enriched zinc ions in the NOX5 sample compared to buffer control (Fig. 4c). We then explored the role of the CXXC motif on NOX5 stability and activity by mutagenesis. Mutation of either cysteine (C568S or C571S) at the zinc finger could destabilize the NOX5 dimer (Fig. S5d) and lead to a diminished enzymatic activity of NOX5 (Fig. 4d). Additionally, addition of TPEN, a zinc chelator, reduced NOX5 activity in a dose-dependent manner (Fig. 4e). These data suggest the CXXC motif at the dimer interface forms a zinc finger, which is important for NOX5 protein stability and kinase activity.”

Further, as the cited literature suggests that zinc inhibits NOX5 activity, how do the authors fit this published data with the proposal that the zinc-binding site is required for oligomerization and activity (vs inhibitory either directly or through the Hv1 channel)?

Thanks for the comments. We clarify here that the inhibitory effect of zinc reported in the literature and the role we propose here are two different molecular events operating in different cellular contexts and zinc concentrations.

Previous literature shows that **extracellular** zinc ions inhibit the Hv1 channel, which blocks proton efflux coupled with the NOX5 catalytic activity, with an IC₅₀ at the micromolar concentration range (PMID: 16554753).

In our study, we observed that zinc ions bind to the **cytosolic** loops of NOX5 and are important for the structure and activity of NOX5 (Fig. 4a-e). Thus, the zinc ion in NOX5 is more like an obligatory metal cofactor. We note that zinc fingers usually have a high affinity for zinc (31 +/- 14 pM); considering that the cellular free zinc concentration is 10-100pM (PMID: 29088067).

A wider discussion of the NOX5 literature, in terms of structure-function and regulatory mechanisms, is needed. Some examples include:

Kawahara et al 2011 showed that oligomerization occurred through the DH domain and that the functional oligomer may be a tetramer (ref 1, see below)

Thanks for the comments. Indeed, we find that NOX5 is fully active as dimers, whereas Kawahara *et al.* suggested that NOX5 forms homo-tetramers in cells, but that was based on the premise that no other protein participates as an essential component of the complex in cells. It is possible that NOX5 forms complexes with unidentified partners in cells. We acknowledge the discrepancy in our manuscript without commenting on potential causes (lines 88-89).

“NOX5 forms homodimers rather than homotetramers as previously reported²³.”

Ref 15 proposes an oxygen binding site, how does this fit with the cryo-EM model presented here?

Thanks for the comments. The oxygen-reacting center has been discussed in previous studies (e.g.: PMID: 32929281 and PMID: 28607049). The putative oxygen binding site is highly conserved in all NOX members, and our data here do not provide any new insights.

DH domain interactions with Hsp90 have been shown to be a crucial regulatory mechanism. How does this fit with the EFD DH domain dynamics, oligomerization and proposed conformational changes? (refs 2-4, see below)

Thanks for the comments. Our study provides no new insights into NOX5 modulation by HSP90, so we prefer not to discuss this in the manuscript.

How do the structural flexibility and dynamics of the EF domain compared to that seen in Fananas et al 2019, in which they see the csNOX5 EFs are partially unfolded in the absence of Ca, and identify conserved aspartates that may be important for DH-EF domain interactions? (ref 5, see below):

Thanks for the comments. Indeed, we find that the EFD is detached and more flexible in the presence of Ca²⁺, but we cannot tell whether it is unfolded.

The two conserved Asp residues (D639 and D649) in csNOX5 identified by Fananas et al. as important for DH-EFD interaction, are located in the REFBD region of human NOX5 (corresponding to D638 and D658). Our structure shows that REFBD mediates the DH-EFD interaction, an observation that is fully consistent with those from Fananas *et al.* We have amended text (lines 118-121) and figure Fig S4h-i to discuss those similarities.

“Consistent with previous work²⁵ showing that two conserved Asp residues (D639 and D649) in csNOX5 are important for DHD-EFD interaction, the corresponding two residues in the human NOX5 (D638 and D658) are located at REFBD (Fig. S4h-i).”

Other concerns:

As oligomerization is thought to be required for activity, details of the dimer interface will be of interest to the field. Therefore, details of the dimer interface should be explained further. The first interface is only described in one sentence (line 124), and the corresponding figure, S5a doesn't make it easy to determine the protomer:protomer interactions. For example, are the Arg residues at the interface? If so, which amino acids are they interacting with? If this is unknown because the FBD: NBD linker is unstructured (and therefore invisible to the cryo-EM), this should be explained. A brief discussion and a figure in which the two protomers are shown in different colors, and the potential interacting amino acids highlighted would be helpful. This goes for site #2 as well, by coloring the protomers in different colors, the readers would be able to easily see the interface.

Thanks for the comments. We have revised Fig. S5a to address the reviewer's comments. Specifically, two protomers are shown in different colors, to allow better visualization of the interface residues. The side chains of potential interface residues in Fig. S5a are not well resolved except for H424. We think that several residues (including F422, H424, R530 and R531) are likely involved in protomer-protomer interaction, but we are not confident about the exact interaction details at this resolution. Moreover, interface #1 is less extensive compared to interface #2. We have also revised the manuscript text (lines 127-130).

“Interface residues may include F422, H424, R426, R530, and R531, but their side chains are not well-resolved for detailed analysis. This interface is likely not as essential as the secondary interface for oligomerization, as point mutations (R426A, R530A or R531A) did not seem to disrupt NOX5 dimers (Fig. S5c).”

Ideally, if this the real dimer interface, mutation of the residues (e.g. Arg) would clearly disrupt it.

Thanks for the comments. We mutated interface #1 residues (R426A, R530A and R531A), and those did not disrupt NOX5 dimerization. The double mutation R530A/R531A seems to affect NOX5 expression and stability and could not be purified. We think interface #1 may be less important than interface #2 for NOX5 oligomerization. We revised the text and Figure S5c (lines 127-130).

“Interface residues may include F422, H424, R426, R530, and R531, but their side chains are not well-resolved for detailed analysis. This interface is likely not as essential as the secondary interface for oligomerization, as point mutations (R426A, R530A or R531A) did not seem to disrupt NOX5 dimers (Fig. S5c).”

Abstract, line 15, there is no need for the “The” before NADPH oxidase 5.

Thanks, we have fixed that.

Intro line 48, this should be re-worded to be more precise. As stated in line 46, Ca²⁺ binding activates DUOX1/2 and NOX5, the difference, as determined by the 2 references stated, is that Ca binding to DUOX relieves an EFD-DH autoinhibitory interaction while Ca binding to NOX5 relieves a DH-DH (REFBD) autoinhibitory interaction. Additionally, this does not necessarily fit with the extensive EFD-DH interactions seen in the cryo-EM model of DUOX1 in the presence of high Ca levels (ref 11).

Thanks for the comments. We have revised lines 45-46 as follows:

“However, activation mechanisms are not conserved between DUOX1-2 and NOX5 due to the different roles of EFD during activation^{15,16}.”

The structure-function relationship between Ca²⁺-bound and Ca²⁺-free DUOX1 (PMID: 33420071) is still puzzling to us, so we chose not to discuss this part in this manuscript.

All the figures should be larger.

Thanks for commenting on the presentation style. We have provided high-resolution and individual larger images for the final submission to ensure a clear visualization of all information in the published paper.

More details are needed for Fig 1a. Is this the final SEC with Superose 6? Is this the Ca-bound or Ca-free prep? Which fractions were collected (and correlate to the SDS page inset)? What is the presumed oligomeric state at this step?

Thanks for the comments. Fig. 1a shows the SEC with Superose 6 increase column for the Ca²⁺-free prep, and the SDS-PAGE inset correlates to the combined peak fractions before grid freezing. We have added the above information to the figure legend (lines 411-413).

“a. Purification of human NOX5 in the Ca²⁺-free condition using a Superose 6 increase column. Right: SDS-PAGE of purified full-length human NOX5 with the peak fractions combined.”

The equations used for the determination of Km, Kcat, and IC50 should be reported (either in results, methods, or figure legend).

Thanks. We used Michaelis-Menten equations for Km and Kcat determination, and the Hill Slope fittings for dose-response curves. This has now been described in the revised Methods session (lines 316-320).

Fig 1: the labeling in f should be made darker (it is so light, especially the yellow and orange, it is hard to read)

Thanks. We have changed the labels in Fig. 1f to make them more readable.

The green vs. teal is very hard to distinguish (Fig, 2e, and S4f)

Thanks for the comments. We have changed the colors to pale green and deep salmon in the corresponding figures (Fig. 2e and S4f).

Fig S3b, please state how the secondary structure predictions were determined (in text, figure legend, or methods)

Thanks for the comments. We used the Jpred4 server (www.compbio.dundee.ac.uk/jpred/), and have cited the paper ([doi:10.1093/nar/gkv332]). The information and citation have been added to the figure legend (line 484).

The text in the results section and Figs 2 and 3 are not well aligned. After Figs 1, S1 and S2, Fig 2c is referenced on lines 94-95, then 3a on lines 100 and 102, 2c-d on line 108 and 120, Fig 4a on line 125 and finally 2a-e on line 143. It would be helpful for the figures to be in the order in which they are referenced in the text.

Thanks for the comments. We have re-arranged Fig. 2 panels to match the order in the text and avoid confusion. We also removed the callout for “Fig. 4a” (line 131).

There does not seem to be a movie as referenced in the text.

We are sorry for the oversight. We have now provided the Movie S1 in the revision.

The csNOX5 and DUOX1 structures appear to have a different orientation of the preTM vs. the NOX5 model (Fig 2f v 2g). Do the authors think this is significant?

Thanks for the comments and apologies for the confusion. Fig S2f and S2g are in slightly different orientation, to better illustrate the lipids. To avoid this confusion, we have added a new panel (Fig. S2h) to show the pre-TM helix orientation. The only structure that has a significantly different orientation is the post-reaction state.

“Grabs” is perhaps not the most precise word for the REFBD:EF interaction and associated conformational changes (lines 144 and 203)

Thanks. We used “displaces” to replace “grabs” in the revision (lines 150 and 215).

WST1 assay for formazan production. Are these all from the same protein prep, or multiple protein preps? For Fig. S4g, is this a representative trace or an average of multiple biological replicates?

Thanks for the comments. We used the same protein prep for replicates within the same assay. For different assays, we use different protein preps. For Fig. S4g, we plot an average of 5 technical repeats (with five aliquots of proteins from the same prep). We have added the following information in the figure legends (lines 490-491).

“Curved are plotted (average +/-SD) with 5 technical replicates.”

Line 157: “We ask how conformational changes in EFD upon Ca²⁺ binding lead to NOX5 activation.” Could be re-phrased.

Thanks for the comments. We have revised the text as follows (lines 163-164):

“We then explored the structure-function relationship between Ca²⁺-induced conformational changes of EFD and NOX5 activation.”

Line 181, “In” is not needed before “consistent with the simulation...”

Thanks, this has been fixed (line 187).

Line 273; NOX5 conc was determined by measuring heme, but NOX5 is not always heme saturated in the cell (4) (unless excess heme was supplied during expression). Then in line 315,

it is stated that conc of NOX5 was determined by OD280. So which assays/experiments used the quantification done with heme?

Thanks for the comments. The reviewer's point is valid, and we'd like to clarify that our HEK cell cultures were supplemented with >2% FBS, which contains heme.

We used more precise heme quantification to quantify the active protein, across different mutations/conditions in our functional assay, based on the fact that only heme-containing proteins are active. In our structural analysis, "intact" NOX5 could be sorted out *in silico* during data processing, providing an estimate of active NOX5.

Line 288, extinction coefficient should have units

Thanks, this has been fixed (line 309).

Were all activity assays conducted at room temp? it only specifically states for calcium conc series.

Yes. We clarified this in the revised text (lines 304 and 321).

The font changes in lines 327-339

Thanks for noticing. We have fixed that and kept font consistent throughout the text.

Abbreviations: FSC, FSEC, should be defined. Alternatively, since it is referred to simply as tryptophan fluorescence in Fig. S5c, the abbreviation FSEC may not be needed at all.

Thanks for the comments. FSC has been defined (line 463) and we revised the Trp fluorescence-based gel filtration chromatography section in Methods as follows (lines 295-302):

"NOX5 mutations (R426A, R530A, R531A, R530A/R531A, C568S, C571S and C568S/C571S) were purified with the same protocol as NOX5 wild type. The GFP tag was cleaved by incubating beads at 4°C with preScission protease for 2 hrs. Flowthrough protein was concentrated and centrifuged for 15 mins with 20,000 g to remove aggregations. The supernatant was applied to a Superose 6 Increase column using the 20 mM Tris pH 8.0, 200 mM NaCl and 0.004% LMNG/CHS (8:1 mass ratio) buffer as mobile phase. Tryptophan fluorescence was measured at the excitation wavelength of 280 nm and emission wavelength 350 nm."

Introduction, lines 31-33 are missing references; Banfi 2001 showed NOX5 expression in testis, spleen and lymph nodes, Bedard, 2012 is a review citing expression of NOX5 in the following tissues; spleen, testis, placenta, uterus, ovary, lymph nodes, pancreas and cells; endothelial cells, VSMCs, cardiac fibroblasts.

NOX5 expression in oligodendrocytes (driving oligodendrocyte differentiation) was shown in 2016 by Acetta et al (ref 6, see below) NOX5 expression in cardiomyocytes was shown in 2012 by Hahn et al. (ref 7, see below) This is not a comprehensive list of tissues and cell types in which NXO5 has been identified, but does cover the tissues and cells listed by the authors.

Thanks. We have revised the introduction to cover the expression of NOX5 in different tissues with references suggested (line 32).

Phenix needs a ref (line 341) Liebschner et al 2019.

Thanks. The reference has been added for Phenix (line 398).

Line 344, please add the pdb(s) and citation.

Thanks. PDB codes (5O0T and 5O0X) and references have been added (line 401).

References 11 and 24 are the same paper

Thanks for catching that. We have now fixed it.

References

1. Kawahara T, Jackson HM, Smith SM, Simpson PD, Lambeth JD. Nox5 forms a functional oligomer mediated by self-association of its dehydrogenase domain. *Biochemistry*. 2011;50(12):2013-25. Epub 2011/02/16. doi: 10.1021/bi1020088. PubMed PMID: 21319793; PMCID: 3073450.
2. Chen F, Pandey D, Chadli A, Catravas JD, Chen T, Fulton DJ. Hsp90 regulates NADPH oxidase activity and is necessary for superoxide but not hydrogen peroxide production. *Antioxidants & redox signaling*. 2011;14(11):2107-19. doi: 10.1089/ars.2010.3669. PubMed PMID: 21194376; PMCID: 3085945.
3. Chen F, Haigh S, Yu Y, Benson T, Wang Y, Li X, Dou H, Bagi Z, Verin AD, Stepp DW, Csanyi G, Chadli A, Weintraub NL, Smith SM, Fulton DJ. Nox5 stability and superoxide production is regulated by C-terminal binding of Hsp90 and CO-chaperones. *Free radical biology & medicine*. 2015;89:793-805. doi: 10.1016/j.freeradbiomed.2015.09.019. PubMed PMID: 26456056.
4. Sweeny EA, Schlanger S, Stuehr DJ. Dynamic regulation of NADPH oxidase 5 by intracellular heme levels and cellular chaperones. *Redox biology*. 2020;36. doi: 10.1016/j.redox.2020.101656.
5. Millana Fananas E, Todesca S, Sicorello A, Masino L, Pompach P, Magnani F, Pastore A, Mattevi A. On the mechanism of calcium-dependent activation of NADPH oxidase 5 (NOX5). *FEBS J*. 2020;287(12):2486-503. Epub 20191220. doi: 10.1111/febs.15160. PubMed PMID: 31785178; PMCID: PMC7317449.
6. Accetta R, Damiano S, Morano A, Mondola P, Paternò R, Avvedimento EV, Santillo M. Reactive Oxygen Species Derived from NOX3 and NOX5 Drive Differentiation of Human Oligodendrocytes. *Frontiers in Cellular Neuroscience*. 2016;10. doi: 10.3389/fncel.2016.00146.
7. Hahn NE, Meischl C, Kawahara T, Musters RJ, Verhoef VM, van der Velden J, Vonk AB, Paulus WJ, van Rossum AC, Niessen HW, Krijnen PA. NOX5 expression is increased in intramyocardial blood vessels and cardiomyocytes after acute myocardial infarction in humans. *The American*

journal of pathology. 2012;180(6):2222-9. Epub 2012/04/17. doi: 10.1016/j.ajpath.2012.02.018.
PubMed PMID: 22503554.

Reviewer #2 (Remarks to the Author):

The manuscript entitled 'Structural Basis of Human NOX5 Activation' by Cui and coworkers used cryo-EM, mutagenesis, and molecular dynamics (MD) simulations to characterize the structures of human NOX5 and propose a working model for NOX5 activation. The study presents valuable insights into the molecular mechanisms of NOX5 activation. Overall, the paper is well-structured and the figures are informative. However, there are several areas that require clarification and expansion for a comprehensive understanding.

We sincerely appreciate the positive comments from the reviewer.

The paper lacks sufficient technical details regarding the cryo-EM experiments, including describing the resolution of the different state models in the results section. Furthermore, how does the local resolution and features change when the models are computed without C2 symmetry?

Thanks for the comments. We have expanded the results/methods sessions to provide that information.

- We stated the overall resolutions in the text (lines 80-85).

"To understand the Ca²⁺-dependent activation of NOX5, we determined cryo-EM structures of NOX5 in different catalytic stages: pre-reaction state (NADPH-bound without Ca²⁺), intermediate states (in the presence of NADPH and Ca²⁺), and post-reaction state (with NADP⁺ and Ca²⁺) (Fig. 1e, S1 and Table S1) at overall resolutions of 3.2 Å, 3.3 Å and 4.1 Å, respectively, with C2 symmetry imposed. Focused refinement followed by symmetry expansion was performed to improve the local resolution of cytosolic domain of NOX5 in the pre-reaction state (see Methods)."

- the resolution of EFD of NOX5 in IS1 and IS2 states was not sufficient for model building. Resolution estimation is not accurate at low resolution, so we didn't put the local resolution maps to avoid confusion.

We revised the text to make that clear, lines 136-142.

"To analyze the EFD motion, we performed 3D classification and refinement without applying symmetry (Fig. 1e-f and S1a) and identified at least three distinct conformations (intermediate state 1–3) with large spatial displacement. The local resolutions of EFD in intermediate state 1 and 2 are insufficient for modeling secondary structure, whereas the intermediate state 3 was resolved to an overall resolution of 3.9 Å, enabling the model building of EFD (Fig. S1a). These conformations, with the EFDs manifesting a trajectory towards the membrane, could represent its motions during NOX5 activation process (Fig. 1e-f and Movie S1)."

And line 428: "All EFD are low pass filtered to 12 Å for comparison between different states."

In addition, we also added more details on the structural determination protocol in the Methods section (lines 392-398).

"For pre-reaction state, symmetry expansion followed by focused refinement was performed to improve the local resolution of cytosolic domains. Mask surrounding DHD and EFD was generated following the Cryosparc tutorial (<https://guide.cryosparc.com/processing-data/tutorials-and-case-studies/mask-selection-and-generation-in-ucsf-chimera>). For intermediate states, a heterogeneous refinement followed by non-uniform refinement was performed to classify out different states without applying C2 symmetry. Local resolution was estimated using half maps as inputs in ResMap software⁵⁰. Model validation was done in Phenix⁵¹.

”

For the focus refinement analysis, how were the masks created? What is the local resolution of the ligands and their environments?

We have described the mask generation protocol in the method session (lines 392-397).

“For pre-reaction state, symmetry expansion followed by focused refinement was performed to improve the local resolution of cytosolic domains. Mask surrounding DHD and EFD was generated following the Cryosparc tutorial (<https://guide.cryosparc.com/processing-data/tutorials-and-case-studies/mask-selection-and-generation-in-ucsf-chimera>). For intermediate states, a heterogenous refinement followed by non-uniform refinement was performed to classify out different states without applying C2 symmetry.”

To show the map quality surrounding ligands, we now show the local cryo-EM density near NADPH of pre-reaction state, IS3 state and intermediate consensus state (Fig. S7e-g).

Details of MD simulations: the authors say that the structure of the system was minimized and equilibrated using ‘standard CHARMM-GUI’ protocol. More details should be added to describe the MD simulations because these protocols are not necessarily known by all potential readers and can be changed by the CHARMM-GUI developers at any time.

Thanks for the comments. These details have been added to the Methods section (lines 351-365).

“The system was then energy minimised and equilibrated using the standard six steps CHARMM-GUI equilibration protocol. This includes the following set-up: The protein backbone was restrained at the force constant of 4000, 2000, 1000, 500, 200 and 50 kJmol⁻¹nm⁻², the protein side chain was restrained the force constant of 2000, 1000, 500, 200, 50, and 0 kJmol⁻¹nm⁻², the lipids non-H atoms were restrained at the force constant of 1000, 400, 400, 200, 40 and 0 kJmol⁻¹nm⁻² and the dihedral restraint was set at the force constant of 1000, 400, 200, 200, 100 and 0 kJmol⁻¹rad⁻². The simulations were equilibrated with a 1 fs timestep for 125 ps for the first three steps, and then to a 2 fs timesteps for 500 ps the next two, and 5 ns for the final step. The first two steps were conducted with the NVT ensemble where the last four were conducted with the NPT ensemble. All equilibration runs were conducted at 310 K using thermostat. In all NPT ensemble equilibration, the pressure was maintained at 1 bar using Berendsen barostat. The production runs were conducted for 500 ns under 310 K using the v-rescale thermostat⁴². The pressure of all systems was maintained at 1 bar using a C-rescale barostat⁴³. All simulations were carried out in triplicates, where simulation frames were saved every 0.1 ns.”

Also, the authors do not provide any detailed analysis to assess if the simulations were well equilibrated after 500 ns. Furthermore, which fraction of these 500 ns was used, for example, for figure 3.e?

Thanks for the comments. All fractions of the 500 ns were used in the analysis. The equilibration and the convergence of the simulations were analyzed using block analyses (Fig. S9). To do so, we analyzed the behavior of our simulations every 100 ns. We showed that the ensemble of the behavior remains consistent after 400 ns, and thus, the 500 ns simulation time is valid.

No details are given on how the electrostatic potentials in fig. 3 and S8 were obtained. Some methods can have serious pitfalls, so explicitly citing which approach was used would help the reader interpret these figures.

Thanks for the comments. We used the PyMOL-Plugin APBS Electrostatics with default setting. We have now clarified this in our method session with APBS referenced (lines 406-408).

“The electrostatic representation in all figures was calculated using PyMOL-APBS Plugin⁵⁵. Calculations were performed at 0.15 M ionic strength in monovalent salt, 298.15 K, protein dielectric 2, and solvent dielectric 78.”

The authors postulate that the ‘flexibility of the nicotinamide group could shorten the distance between NADPH and FAD and thus trigger electron transfer’. However, they do not provide a molecular rationale in which the local environment of the nicotinamide group is explored in the structures and/or MD simulations. How do the coordination of the different groups change in the different states?

Thanks for the comments. The major difference between the states is the position of pre-TM helix (Fig. S7b), resulting in differences in the surface charge potential between pre-reaction and IS3 (Fig. 3c). We have added a supplemental figure to show the difference in the pocket (Fig. S7e-g). However, due to the limit in local resolution of pre-TM (the part that changes among different states), especially in the IS3, we prefer not to discuss the potential implications in further atomic details. We also show the RMSD changes of different chemical groups in Fig. S9a during the simulations.

Fig 1.a: provide more detailed legend for the left panel.

We have added more information to the figure legend (lines 411-413).

“a. Purification of human NOX5 in the Ca²⁺-free condition using a Superose 6 increase column. Right: SDS-PAGE of purified full-length human NOX5 with the peak fractions combined.”

Fig 1e: indicate which domains encompass the DH.
This was done.

Fig 1.f: this panel is unclear to me. Please indicate what the top and bottom panel represent.

We apologize for the lack of clarity. We have explained the panels in the revised legend as follows (lines 422-428):

“f. The different conformations of EFD in two views (side and bottom). Top and bottom panel represent side and bottom view, respectively. Here, the catalytic modules of different states from Fig. 1e are aligned to the consensus map of the intermediate state (white). Micelles are shown to indicate the membrane position, and only the consensus map of the intermediate state are shown for clarity. Different conformations of EFD are colored in dark grey (pre-reaction), violet (IS1), yellow (IS2), orange (IS3) and green (post-reaction). IS1-3 indicate intermediate state 1-3. All EFD are low pass filtered to 12 Å for comparison between different states.”

Fig 2.b: please indicate which helix is shown in black.

It is the REFBD motif. We clarified this point in the figure legend (lines 434-436).

“d. Distance change between C-terminal of EFD and preTM1 in pre-reaction state and intermediate state 3. The REFBD motif is colored in black.”

Fig 2: It is difficult to visualize the statement ‘EF3-4 opens up, grabs the REFBD motif and detaches it from the NBD’. In particular, it is not clear if there is a binding effect or if the REFBD conformational change is just a consequence of the connectivity of the protein. Is it really that the REFBD protein-protein interface is changing from EF3-4 to the NBD?

We are not sure we fully understand the question here, but we acknowledge our description was not clear. In the pre-reaction state, EFD-REFBD-NBD interact (Fig. 2a and S4d). Upon Ca^{2+} binding, EF3-4 opens up, REFBD sits in the groove between EFD3-4 (Fig. 2b and S4e) and moves away from NBD (Fig. 2a and S4d). We now illustrate these conformational changes in Movie S1.

Fig 3: I would suggest using the same naming and order when referring to the different states. In this figure using Ca^{2+} -free state, Ca^{2+} -bound state, intermediate state-3, Ca^{2+} , and EGTA is confusing about the equivalence between states.

We appreciate the comment and suggestion. We now use "pre-reaction state" instead of Ca^{2+} -free state and "IS3 state" instead of Ca^{2+} -bound state in Figure 3. We still use the terms Ca^{2+} -bound and Ca^{2+} -free for DUOX1, as we don't have the corresponding post-reaction state from the literature.

In conclusion, the paper offers valuable insights into the molecular mechanisms of NOX5 activation and its structural characteristics. To enhance its clarity and impact, the paper should provide more straightforward explanations of the experimental and computational methods. Providing more technical details would strengthen the paper's overall quality.

We appreciate the constructive comments. We find that the revisions we made to address those comments have greatly improved and strengthened our paper.

Reviewer #3 (Remarks to the Author):

NADPH oxidase 5 (NOX5) of the NOX family catalyzes production of superoxide by transferring electrons from cytosolic NADPH to extracellular oxygen. Its enzymatic activity is regulated by multiple intracellular factors, such as Ca^{2+} , Zn^{2+} , and phosphorylation. Ca^{2+} is required for the activation of NOX5; however, its activation mechanism is unclear due to the lack of high-resolution structural information. This manuscript by Cui and coauthors reports the cryo-EM structures of NOX5 in three different states: pre-reaction, intermediate, and post-reaction states. By interpreting these structures and supported by mutagenesis analyses and MD simulations, the authors propose a mechanism for NOX5 activation by Ca^{2+} that the binding of Ca^{2+} in the EF-hand domain (EFD) leads to motions of the EFD and increases dynamics of NADPH, therefore allowing electron transfer from NADPH to FAD. This is an intriguing new discovery that will be an important step to advance our understanding of the mechanisms of activation/regulation and electron transfer in NOX5 and more broadly in the NOX family.

We sincerely appreciate the positive comments of the reviewer.

Specific comments:

1. The methods for cryo-EM sample preparation of NOX5 in the three different conditions are not clearly described. It is useful and important to provide sufficient details, such as the concentration of each key compound, incubation time, etc.

Thanks for the suggestion. We have provided the experimental details in the Methods section as follow (lines 368-377):

“For pre-reaction state cryo-EM sample preparation, purified NOX5 (~8 mg/mL by OD280) was supplemented with EGTA and NADPH to final concentration of 5 mM and 1 mM, respectively. For the intermediate state sample, purified NOX5 was supplemented with CaCl_2 and NADPH to final concentration of 1 mM. Cryo-EM grids were prepared by adding 3.5 μL sample to the Quantifoil R1.2/1.3 holey carbon gold grids using a Vitrobot Mark IV system (FEI). Freezing was performed with a blot force of -3, a blot time of 3 s, and a wait time of 15 s under 100% humidity. NOX5 post-reaction state sample was prepared with EM GP2 (Lecia). Purified NOX5 (~10 mg/mL by OD280) was supplemented with CaCl_2 , NADP⁺ and FAD to final concentration of 1 mM, 1.2 mM and 60 μM , respectively. The mixture was incubated for half an hour before freezing grids. 3.5 μL sample was applied and frozen with a blot time of 4 s, and a wait time of 5 s under 95% humidity at 8°C.”

2. The authors did meticulous analyses of the cryo-EM data and found populations of particles in different conformations. However, they did not discuss possible variations caused by cryo-EM grid preparation. There are a few illy controlled factors in cryo-EM grid preparation, such as interactions of protein with the air/water interface, ice thickness, increase of salt/detergent concentration caused by evaporation during grid blotting, etc. The reviewer suggests the authors repeat their cryo-EM experiments to confirm that conformational changes/motions are truly from Ca^{2+} binding and not other unrelated variations.

Thanks for the comments. The structures reported here were not from a single experiment. In fact, for structural determination, we performed numerous optimization trials (grid types, freezing conditions, etc.), which yielded many low-resolution maps. These maps, though not appropriate

for model building, give us high confidence that EFD flexibility we observed was due to Ca^{2+} supplementation.

3. The 2D class averages in Fig. S1 show dimers that vary on the EFD. Some have two EFDs and others have only one. It is interesting that this difference is orientation-dependent. The reviewer is concerned that one of two EFDs interacts with the air/water interface and becomes mobile or unfolded. It is also possible that the EFD is invisible at some angles when it is overlapping with the TMD. It is a concern that the authors obtained asymmetric structures of the intermediate states with only one EFD, although the 2D class averages show a good 2-fold symmetry and both EFDs in the dimer.

Thanks for the comment.

We disagree with the reviewer's statement that some 2D classes have only one EFD. This incorrect impression may result from the specific projection angles; as the reviewer notes, the apparent difference is orientation-dependent. Below, we compare the experimental 2D classes with 2D projections of C2-symmetry imposed map to show that the 2-fold symmetry map, due to different projection angle, can seem to have only one EFD.

2D classes of the pre-reaction (experimental)

2D projections of the C2-symmetry imposed NOX5 (pre-reaction state)

For the Intermediate state sample, there are also two EFDs. The EFD in this state is “fussier” than the other two states due to flexibility. However, if we lower our threshold (0.169 in Chimera), we could see densities for both EFDs. We filtered them to similar levels for better comparison.

From left to right: intermediate state 1, intermediate state 2 and intermediate state 3. EDFs are indicated by dashed circles.

4. The cryo-EM density of the Ca^{2+} bound intermediate state 3 shows weak density of the nicotinamide moiety of NADPH. The authors suggest this is caused by an increased flexibility of this moiety. But this may also be a result of the lower resolution of this structure. Also, the consensus structure of the Ca^{2+} bound state seems to show a strong density of the nicotinamide moiety (Fig. 2f).

Thanks for the comments. We initially shared the reviewer's concern that the lower NADPH density in IS3 could be due to flexibility or lower resolution. However, the following observations lead us to think the flexibility of NADPH is the main factor here.

- 1) We get an overall higher resolution with the pre-reaction state than the intermediate and post-reaction states.
- 2) The protein densities near NADPH in different states are reasonably well resolved (see Fig. S7e-g)
- 3) In the same supplemental figures (see Fig. S7g), we added the density map of NADPH from the consensus structure, which also showed weaker density of the nicotinamide group.

We are thus confident in our interpretation. We have added Fig. S7g to make that point clear.

5. The authors should show the densities of both NADPH and NOX5 in Fig. 3d for a more reliable interpretation of the cryo-EM map.

Thanks for the suggestion. We show in Fig. S7e-g densities of both NADPH and surrounding domains.

6. The reviewer is not an expert on MD simulations, but feel cautious about the simulations results in the paper because the models used in the simulations contain a few low-resolution domains and missing loops/side chains that are modelled using SwissModel.

We thank the reviewer for the concern. We used SwissModel to model the missing loops simply to prevent any alteration from the cryo-EM structure. Thus, what is not seen in the structures was not simulated, and the loops were then artificially connected as described in the Methods section. Moreover, our RMSD analysis (Fig S9c-d) highlights that all sidechains within the protein structures are stable throughout all repeats, giving us further confidence about the simulation results.

7. The conclusion about the Zn-binding motif is speculative because neither a clear density of Zn^{2+} ion is resolved in the cryo-EM maps nor experiments were performed to confirm it is Zn^{2+} and not other ions. It is more precise to call it a putative Zn-binding motif.

Thanks for the comments. We have followed the reviewer's suggestion and changed the subtitle of the result session (line 191) and further explored the zinc-binding site in our revision:

- we detected zinc ions in purified NOX5 using inductively coupled plasma mass spectrometry (ICP-MS), an analytical technique that can measure elements at trace levels in biological samples (Fig 4c).

- we tested NOX5 activity in the presence of different concentrations of TPEN, a zinc chelator, and observed dose-dependent inhibition of NOX5 by TPEN (Fig. 4e).

These two pieces of evidence, combined with the cysteine mutagenesis data (cysteine mutation reduces NOX5 stability and activity and alters oligomerization state), support our interpretation of the cryo-EM density map that a zinc ion is likely present at the dimerization interface, and that it is important for the structure and the enzymatic activity of NOX5. We have thus revised our manuscript accordingly (lines 197-205).

“The presence of Zn²⁺ was supported by the inductively coupled plasma mass spectrometry (ICP-MS) analysis, which showed enriched zinc ions in the NOX5 sample compared to buffer control (Fig. 4c). We then explored the role of the CXXC motif on NOX5 stability and activity by mutagenesis. Mutation of either cysteine (C568S or C571S) at the zinc finger could destabilize the NOX5 dimer (Fig. S5d) and lead to a diminished enzymatic activity of NOX5 (Fig. 4d). Additionally, addition of TPEN, a zinc chelator, reduced NOX5 activity in a dose-dependent manner (Fig. 4e). These data suggest the CXXC motif at the dimer interface forms a zinc finger, which is important for NOX5 protein stability and kinase activity.”

REVIEWERS' COMMENTS

Reviewer #1 (Remarks to the Author):

The authors have been responsive to the concerns listed in the initial review, and I am satisfied by the alterations and additions they have made. Overall, this is a very nice piece of work that is highly relevant for the NOX field and structure-function mechanisms of electron transport.

Reviewer #2 (Remarks to the Author):

The 'Structural Basis of Human NOX5 Activation' manuscript by Cui et al. has been revised to include an expanded methods section, additional evidence supporting Zn binding, new supplementary figures, and addresses of most my concerns. Overall, the paper offers new insights into the structure-function of NOX5. The manuscript is well-structured, the conclusions are well-supported by the data, and I believe it should be accepted for publication.

Minor comments:

The authors do not explain how they used the different MD replicates to validate their results. All figures show single result. Are the figures from one replica or from pooled data?

Is it possible to provide a better citation for 'Mask surrounding DHD and EFD was generated following the Cryosparc tutorial (<https://guide.cryosparc.com/processing-data/tutorials-and-case-studies/mask-selection-and-generation-in-ucsf-chimera>)'? Tutorials can be changed or deleted from the website, making reproducing the results difficult.

Reviewer #3 (Remarks to the Author):

This revised manuscript has properly addressed the reviewer's comments.

POINT-BY-POINT RESPONSE TO REVIEWER #2's COMMENTS

We thank the positive feedback from reviewers #1 and #3.

Reviewer #2 (Remarks to the Author):

The 'Structural Basis of Human NOX5 Activation' manuscript by Cui et al. has been revised to include an expanded methods section, additional evidence supporting Zn binding, new supplementary figures, and addresses of most my concerns. Overall, the paper offers new insights into the structure-function of NOX5. The manuscript is well-structured, the conclusions are well-supported by the data, and I believe it should be accepted for publication.

We sincerely appreciate the reviewer's positive comments.

Minor comments:

The authors do not explain how they used the different MD replicates to validate their results. All figures show the single result. Are the figures from one replica or from pooled data?

Figures are from pooled data. we explained in figure legend (lines 644, 646-647).

"Data in all triplicates are combined to a single histogram."

We also add some details in the methods part (lines 360-363):

"All simulations were carried out in triplicates where velocity is randomly generated at the beginning of every simulation. All simulation frames were saved every 0.1 ns. All data from one condition is combined to a single histogram."

Is it possible to provide a better citation for 'Mask surrounding DHD and EFD was generated following the Cryosparc tutorial (<https://guide.cryosparc.com/processing-data/tutorials-and-case-studies/mask-selection-and-generation-in-ucsf-chimera>)'? Tutorials can be changed or deleted from the website, making reproducing the results difficult.

To clearly describe how we generate the mask surrounding DHD and EFD, we add more details in methods part (lines 393-402):

"Mask surrounding DHD and EFD was generated as follows: refined map was imported into ChimeraX⁵⁰ using volume eraser to remove transmembrane domain, cytosolic domains (DHD and EFD) map (map 1) was saved. Command line (relion_image_handler with the options --lowpass 15 --angpix 0.826) was used to create a low-pass filtered map for cytosolic domains, the low-pass filtered map was opened in ChimeraX to test the threshold (value 1) at which map showed no noisy spots outside the protein area. Then cytosolic domains map (map 1) was input into Mask Creation of RELION^{48,49} with the options: Lowpass filter map (A) of 15, Initial binarization threshold of value 1, Extend binary map this many pixels and Add a soft-edge of this many pixels of 6 to generate the mask of cytosolic domains (DHD and EFD)."